# Astrocytic phagocytosis is a compensatory mechanism for microglial dysfunction

Hiroyuki Konishi[1],[*] (ID), Takayuki Okamoto[1], Yuichiro Hara[2,3], Okiru Komine[4], Hiromi Tamada[1], Mitsuyo Maeda[5,6], Fumika Osako[1], Masaaki Kobayashi[1], Akira Nishiyama[7], Yosky Kataoka[5,6], Toshiyuki Takai[8], Nobuyuki Udagawa[9], Steffen Jung[10] (ID), Keiko Ozato[11], Tomohiko Tamura[7], Makoto Tsuda[12] (ID), Koji Yamanaka[4] (ID), Tomoo Ogi[2,3], Katsuaki Sato[13] & Hiroshi Kiyama[1],[**] (ID)

## Abstract

Microglia are the principal phagocytes that clear cell debris in the central nervous system (CNS). This raises the question, which cells remove cell debris when microglial phagocytic activity is impaired. We addressed this question using *Siglech*[dtr] mice, which enable highly specific ablation of microglia. Non-microglial mononuclear phagocytes, such as CNS-associated macrophages and circulating inflammatory monocytes, did not clear microglial debris. Instead, astrocytes were activated, exhibited a pro-inflammatory gene expression profile, and extended their processes to engulf microglial debris. This astrocytic phagocytosis was also observed in *Irf8*-deficient mice, in which microglia were present but dysfunctional. RNA-seq demonstrated that even in a healthy CNS, astrocytes express TAM phagocytic receptors, which were the main astrocytic phagocytic receptors for cell debris in the above experiments, indicating that astrocytes stand by in case of microglial impairment. This compensatory mechanism may be important for the maintenance or prolongation of a healthy CNS.

**Keywords** astrocyte; debris; microglia; phagocytosis; RNA-seq
**Subject Categories** Neuroscience
**The EMBO Journal (2020) 39: e104464**

## Introduction

Microglia are macrophage-related cells of the central nervous system (CNS). They originate from the yolk sac, invade the parenchyma during development, and reside in the CNS throughout life (Waisman *et al*, 2015). Microglia are regarded as the principal phagocytes in the CNS (Wolf *et al*, 2017); they engulf dying or dead cells depending on the situation. In the developing CNS, a number of unhealthy or misconnected neurons die (Oppenheim, 1991), and their debris are cleared by microglia (Rigato *et al*, 2011). Upon neuronal injury, a large amount of cellular debris accumulates at the injury site, which is scavenged by reactive microglia (Sierra *et al*, 2013). Failure to clear debris has detrimental effects on surrounding neural tissue. For instance, accumulated debris can be a barrier to growing axons (Chen *et al*, 2000; Tanaka *et al*, 2009). More importantly, some intracellular molecules can leak from dead cells and trigger inflammatory responses in surrounding cells, resulting in damage to neural tissue or activation of autoimmunity (Sierra *et al*, 2013). The removal of dying or dead cells by microglia is, therefore, essential for development, maintenance, and regeneration of the CNS. However, impairment of microglial phagocytosis can occur in certain conditions, such as aging and injury (Abiega *et al*, 2016; Pluvinage *et al*, 2019). Alternatively, in the event of brain ischemia or traumatic injury, a large amount of cellular debris can overwhelm microglial capacity (Ritzel *et al*, 2015). In this context, auxiliary or supportive clearance systems may be actuated in the CNS.

1 Department of Functional Anatomy and Neuroscience, Nagoya University Graduate School of Medicine, Nagoya, Japan
2 Department of Genetics, Research Institute of Environmental Medicine, Nagoya University, Nagoya, Japan
3 Department of Human Genetics and Molecular Biology, Graduate School of Medicine, Nagoya University, Nagoya, Japan
4 Department of Neuroscience and Pathobiology, Research Institute of Environmental Medicine, Nagoya University, Nagoya, Japan
5 Multi-Modal Microstructure Analysis Unit, RIKEN-JEOL Collaboration Center, Kobe, Japan
6 Laboratory for Cellular Function Imaging, RIKEN Center for Biosystems Dynamics Research, Kobe, Japan
7 Department of Immunology, Yokohama City University Graduate School of Medicine, Yokohama, Japan
8 Department of Experimental Immunology, Institute of Development, Aging and Cancer, Tohoku University, Sendai, Japan
9 Department of Biochemistry, Matsumoto Dental University, Shiojiri, Japan
10 Department of Immunology, Weizmann Institute of Science, Rehovot, Israel
11 Division of Developmental Biology, Eunice Kennedy Shriver National Institute of Child Health and Human Development, National Institutes of Health, Bethesda, MD, USA
12 Department of Life Innovation, Graduate School of Pharmaceutical Sciences, Kyushu University, Fukuoka, Japan
13 Division of Immunology, Department of Infectious Diseases, Faculty of Medicine, University of Miyazaki, Miyazaki, Japan
   *Corresponding author. Tel: +81 52 744 2015; Fax: +81 52 744 2027; E-mail: konishi@med.nagoya-u.ac.jp
   **Corresponding author. Tel: +81 52 744 2015; Fax: +81 52 744 2027; E-mail: kiyama@med.nagoya-u.ac.jp

To address this alternative clearance system, a microglia-specific ablation model, in which the debris of dying microglia can be tracked in the absence of microglial phagocytosis, can provide important insights. Conditional microglial ablation models using both genetic and pharmacological strategies have been established (Waisman *et al*, 2015; Han *et al*, 2019). However, previously reported ablation systems may not be suitable for identifying a compensatory phagocytosis system because of non-microglial mononuclear cells in the CNS (Prinz *et al*, 2017). CNS-associated macrophages reside in the CNS boundary regions, such as the meninges, perivascular space, and choroid plexus (Goldmann *et al*, 2016). In addition, circulating monocytes infiltrate the CNS upon inflammation or injury (Yamasaki *et al*, 2014). Because these non-microglial mononuclear cells have phagocytic properties (Yamasaki *et al*, 2014; Prinz *et al*, 2017), it is likely that they participate in clearance of microglial debris. However, using previously reported genetic systems, at least some of these cells are assumed to be ablated concomitantly with microglia. This is because mononuclear lineages have similar gene expression profiles, such as for *Itgam* (the gene encoding cluster of differentiation [CD]11b), *Cx3cr1*, and *Aif1* (the gene encoding ionized calcium-binding adaptor molecule 1 [Iba1]; Wieghofer & Prinz, 2016). Using *Cx3cr1*CreER-based genetic ablation systems, CX3C chemokine receptor 1 (CX3CR1)[+] monocytes circumvent death because of their short life span (Goldmann *et al*, 2013; Parkhurst *et al*, 2013; Yona *et al*, 2013; Bruttger *et al*, 2015; Cronk *et al*, 2018; Lund *et al*, 2018); however, most CNS-associated macrophages are assumed to be ablated concomitantly with microglia, given their *Cx3cr1* expression and longevity (Goldmann *et al*, 2016). A current standard pharmacological ablation system uses colony-stimulating factor-1 receptor (CSF1R) inhibitors, PLX3397 or PLX5622 (Elmore *et al*, 2014; Dagher *et al*, 2015). A very recent study showed that PLX3397 almost completely depletes all types of CNS-associated macrophages (Van Hove *et al*, 2019). To improve the specificity of cell ablation, we recently produced a new mouse model utilizing the *Siglech* locus. Studies by ourselves and others demonstrate that sialic acid-binding immunoglobulin-like lectin H (Siglec-H) expression is almost entirely confined to microglia in the CNS; its expression is absent in circulating monocytes and CNS-associated macrophages, except for a fraction of choroid plexus macrophages (Konishi *et al*, 2017; Van Hove *et al*, 2019). Accordingly, *Siglech*dtr mice, in which diphtheria toxin (DT) receptor (DTR) is knocked into the 3′-untranslated region of the *Siglech* gene, enable highly specific ablation of microglia without affecting most other mononuclear populations (Konishi *et al*, 2017).

In this study, we demonstrate that astrocytes, rather than non-microglial mononuclear cells, readily phagocytose microglial debris in our microglia ablation model. Furthermore, astrocytes are capable of engulfing spontaneous apoptotic cells in mutant mice with phagocytosis-impaired microglia, but not in wild-type (WT) mice, indicating that astrocytes have the potential to compensate for microglia with dysfunctional phagocytic activity.

# Results

## Microglial-independent clearance of microglial debris after ablation of microglia

Siglec-H expression is almost entirely confined to microglia among CNS-related mononuclear cells (Fig EV1), and ablation using

*Siglech*dtr mice is nearly specific for microglia (Konishi *et al*, 2017). Here, we focused on the hippocampal CA1 region because it has a high ablation rate for microglia. Intraperitoneal administration of DT induced apoptosis of Iba1[+] microglia, which were identified by pyknotic or fragmented nuclei, in the hippocampal CA1 of *Siglech*dtr/dtr mice (Fig 1A and B). The number of live microglia with normal nuclei was significantly reduced by ~85% in the 2–4 days post-DT injection (Fig 1A and C). The number then recovered to normal levels by day 7, although the repopulated microglia exhibited less ramified morphology (Appendix Fig S1). Thereafter, by day 28, the morphology became identical to that before DT administration.

In previously reported microglial ablation models, the origin of repopulated microglia is controversial (Waisman *et al*, 2015; Han *et al*, 2019); therefore, we first addressed this issue (Fig EV2). Parabiotic coupling of *Siglech*dtr/dtr mice with CAG-EGFP mice, in which green fluorescent protein (GFP) is expressed in all cells under the control of a β-actin-based CAG promotor, resulted in 54.8 ± 3.5% (mean ± SEM, *n* = 5) GFP[+] peripheral blood cells 3 weeks after parabiotic surgery (Fig EV2A and B). Seven days after DT administration, lymph nodes, as a positive control, contained a significant number of GFP[+] cells (Fig EV2C). By contrast, there were no GFP[+] microglia in hippocampal CA1, indicating that repopulated microglia are not derived from circulating blood cells, including any myeloid lineage, which are reported to enter the brain parenchyma and differentiate into microglia-like cells in some microglial ablation models (Lund *et al*, 2018; Shemer *et al*, 2018). On day 4, the onset of the recovery phase, ~90% of microglia expressed proliferation marker Ki-67 in their nucleus (Fig EV2D and E), indicating that survived microglia actively proliferate to replenish the brain, which is similar to other microglial ablation models, such as using *Cx3cr1*CreER:*Rosa26*dtr mice (Bruttger *et al*, 2015) or CSF1R inhibitor PLX compounds (Huang *et al*, 2018; Van Hove *et al*, 2019).

In our ablation model, we found a large amount of microglial debris (Iba1[+] spheres with a diameter > 2 μm and no nuclei) on day 2; however, the debris was diminished by day 4 (Fig 1A and D). During this time, most of residual microglia exhibited abnormal morphology with fewer ramified processes (Appendix Fig S1), and did not interact with the debris (Fig. 1E), indicating that a microglia-independent clearance system worked in the absence of functional microglia.

## Non-microglial mononuclear populations do not participate in debris clearance

Most non-microglial mononuclear cells are not expected to be ablated in this ablation system (Fig EV1); therefore, we explored the possibility that microglial debris was cleared by such cells around day 2. We first considered the involvement of CC chemokine receptor 2 (CCR2)[+]Ly6Chigh inflammatory monocytes (Fig EV1; Serbina & Pamer, 2006), which conditionally infiltrate diseased or inflamed CNS and act as phagocytes (Yamasaki *et al*, 2014; Ritzel *et al*, 2015). As a positive control, we crossed *Siglech*dtr with *Ccr2*RFP mice to label CCR2[+] cells with red fluorescent protein (RFP) and confirmed that a large number of RFP[+] cells infiltrated the spinal cord of *Siglech*dtr/dtr:*Ccr2*RFP/+ mice after induction of experimental autoimmune encephalomyelitis (EAE; Fig 2A). As expected, 2 days after DT administration to *Siglech*dtr/dtr:*Ccr2*RFP/+ mice, RFP[+]Ly6Chigh inflammatory monocytes were present in peripheral blood,

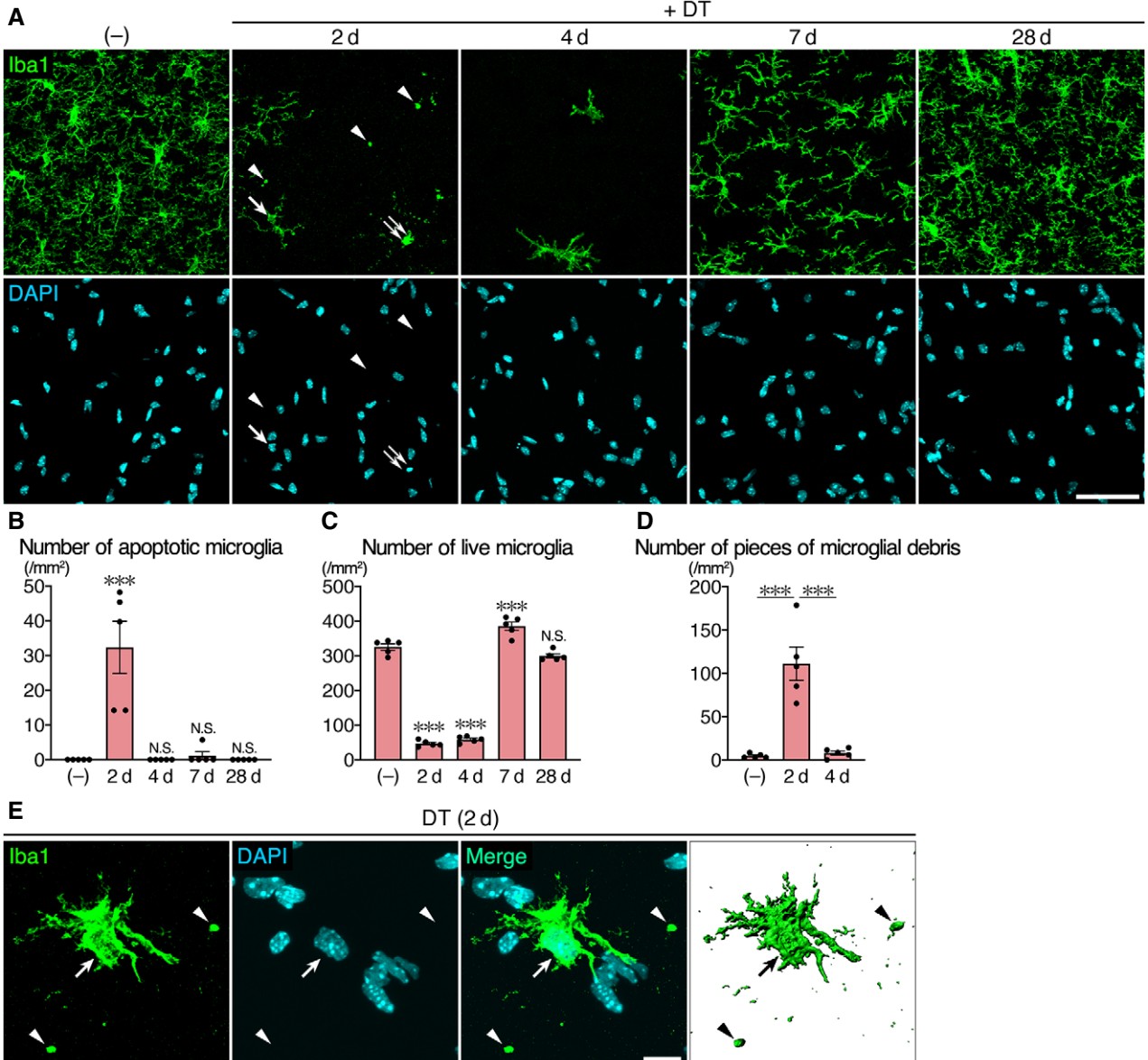

**Figure 1. Clearance of microglial debris after microglial ablation.**

A   Time-course images of the hippocampal CA1 region of *Siglech*dtr/dtr mice after DT administration. Sections were stained with an anti-Iba1 antibody (green) and DAPI (cyan). Arrow: live microglia with normal nucleus. Double arrow: apoptotic microglia with pyknotic nucleus. Arrowhead: microglial debris with no nucleus. Scale bar, 50 μm.

B   Number of apoptotic microglia with pyknotic or fragmented nucleus (n = 5 animals per group, Kruskal–Wallis test with *post hoc* Dunn's test).

C   Number of live microglia with normal nucleus (n = 5 animals per group, one-way ANOVA with *post hoc* Tukey's test).

D   Number of pieces of microglial debris (Iba1+ spheres with a diameter > 2 μm and no nucleus) (n = 5 animals per group, Kruskal–Wallis test with *post hoc* Dunn's test).

E   A representative image of a survived microglial cell in the hippocampal CA1 region of *Siglech*dtr/dtr mice 2 days after DT administration. Sections were stained with an anti-Iba1 antibody (green) and DAPI (cyan). A 3D image (rightmost panel) was reconstructed using Imaris software. Arrow: live microglia. Arrowhead: microglial debris. Scale bar, 10 μm.

Data information: Values show the mean ± SEM. N.S.: no significance; ***P < 0.001.

and their number was slightly increased by an unknown mechanism (Fig 2B). Even in the presence of circulating inflammatory monocytes after DT injection, we found no infiltrated RFP+ cells in hippocampal CA1 2–4 days after DT administration (Fig 2A),

eliminating the possibility that inflammatory monocytes infiltrated the hippocampal parenchyma and cleared microglial debris.

We next focused on CNS-associated macrophages, which are located at boundary regions of the CNS and specifically express

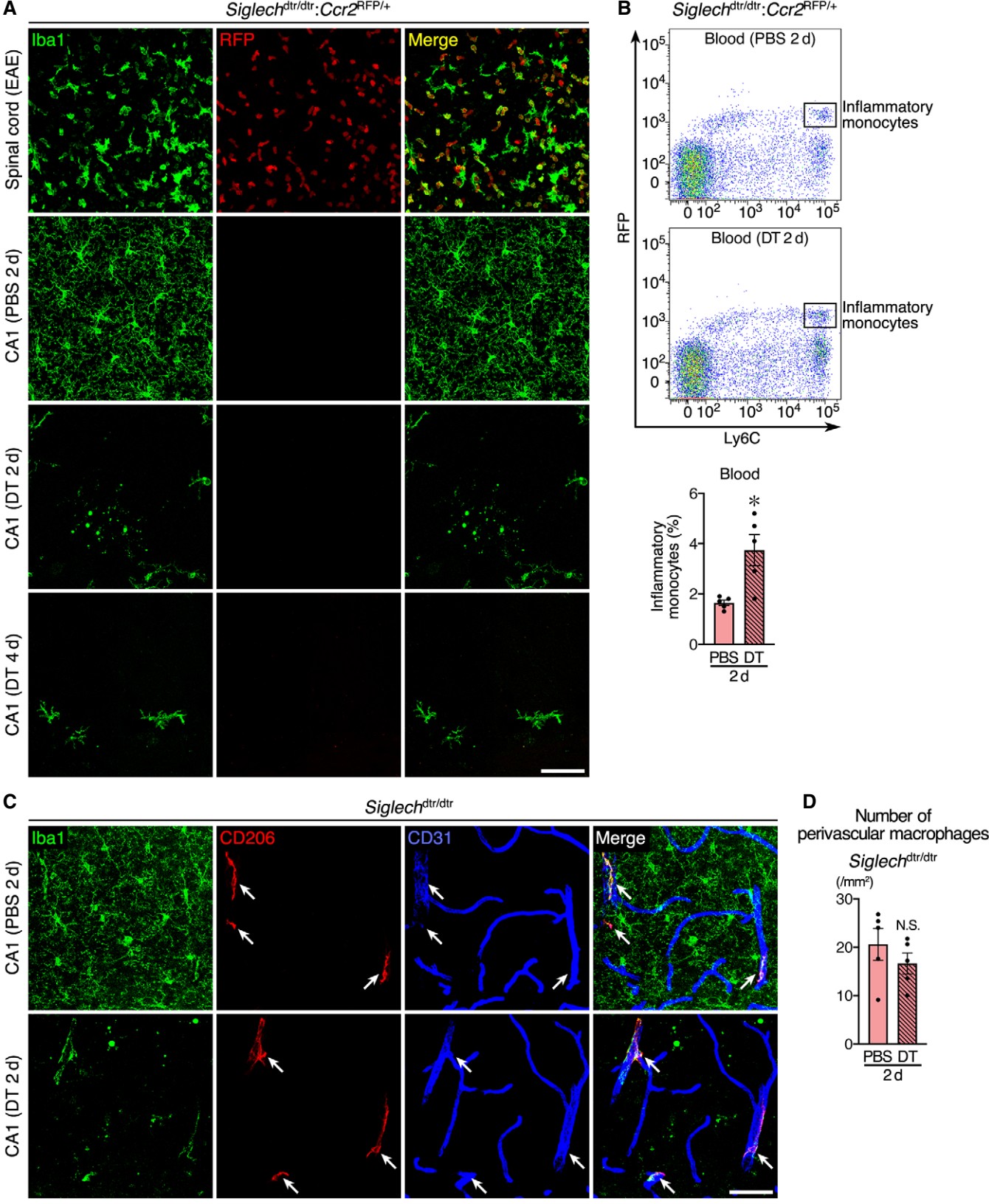

**Figure 2.**

**Figure 2.  Non-microglial mononuclear populations do not act as scavengers for microglial debris.**

A   Histological analysis of RFP⁺ cells in *Siglech*^dtr/dtr^:*Ccr2*^RFP/+^ mice. Lumbar spinal cord after EAE induction and hippocampal CA1 2 or 4 days after PBS or DT injection were analyzed. Iba1 immunoreactivity (green) and RFP fluorescence (red) are shown. Images were acquired using the same laser power and sensitivity, and image processing was the same for all RFP signals (red). Scale bar, 50 μm.

B   Flow cytometric analysis of RFP⁺Ly6C^high^ inflammatory monocytes in peripheral blood of *Siglech*^dtr/dtr^:*Ccr2*^RFP/+^ mice 2 days after PBS or DT administration. Representative data and a quantification graph (*n* = 5 animals per group, two-tailed unpaired Student's *t*-test) are shown.

C   Immunohistochemical localization of CD206⁺ perivascular macrophages (arrows) in hippocampal CA1 of *Siglech*^dtr/dtr^ mice. Sections prepared 2 days after administration of PBS or DT were stained with anti-Iba1 (green), anti-CD206 (red), and anti-CD31 (blue) antibodies. Scale bar, 50 μm.

D   Number of perivascular macrophages in hippocampal CA1 of *Siglech*^dtr/dtr^ mice 2 days after PBS or DT administration (*n* = 5 animals per group, two-tailed unpaired Student's *t*-test).

Data information: Values show the mean ± SEM. N.S.: no significance; *$P$ < 0.05.

CD206 in contrast to microglia (Fig EV1) (Goldmann *et al*, 2016). Among these macrophage cell types, the involvement of leptomeningeal macrophages and choroid plexus macrophages was excluded, given the distance of the leptomeninges and choroid plexus from the hippocampal parenchyma. In contrast to these two cell types, perivascular macrophages were found in association with medium- or large-sized CD31⁺ vessels in the hippocampal CA1 of control mice (Fig 2C). Consistent with our previous result obtained in the cerebral cortex (Konishi *et al*, 2017), perivascular macrophages remained intact even 2 days after DT administration in hippocampal CA1 (Fig 2D). They still associated with vessels, similarly to the PBS-administered mice, and did not infiltrate hippocampal parenchyma for debris clearance (Fig. 2C). Together, these results suggest that brain-resident non-professional phagocytes play roles in debris clearance.

**Activated astrocytes engulf microglial debris**

To investigate the microglia-independent clearance system, we analyzed mRNA levels of marker molecules for various CNS cell types in the hippocampus 2 days after DT treatment (Fig 3A). Expression of *Aif1* (the gene encoding Iba1) was significantly decreased, confirming microglial ablation. Expression of *Map2* (a neuron marker), *Mbp* (an oligodendrocyte marker), and *Cspg4* (the gene encoding oligodendrocyte precursor cell marker, neuron-glial antigen 2 [NG2]) was not altered. By contrast, the expression of astrocyte marker *Gfap* was significantly increased after microglial ablation. Immunohistochemical staining of glial fibrillary acidic protein (GFAP) demonstrated that astrocytes exhibited hypertrophic morphology (Fig 3B), which is a hallmark of astrocytic activation (Sun & Jakobs, 2012), although the number of cells was unchanged (Fig 3C).

These results prompted us to look for interactions between astrocytes and microglial debris. For this purpose, we used an antibody against CD11b, instead of Iba1, because microglial debris was detected more clearly by staining for CD11b than Iba1 (Appendix Fig S2). In contrast to PBS-injected control mice, we frequently observed that astrocytes extended their processes to contact microglial debris 2 days after DT administration (Fig 4A). We also occasionally found that a single piece of microglial debris was surrounded by several processes from multiple astrocytes. The interaction between activated astrocytes and microglial debris was observed throughout the brain, including in the cerebral cortex, thalamus, and medulla (Fig EV3).

We then examined whether this phenomenon is conserved among different microglial ablation models (Fig EV4). In another

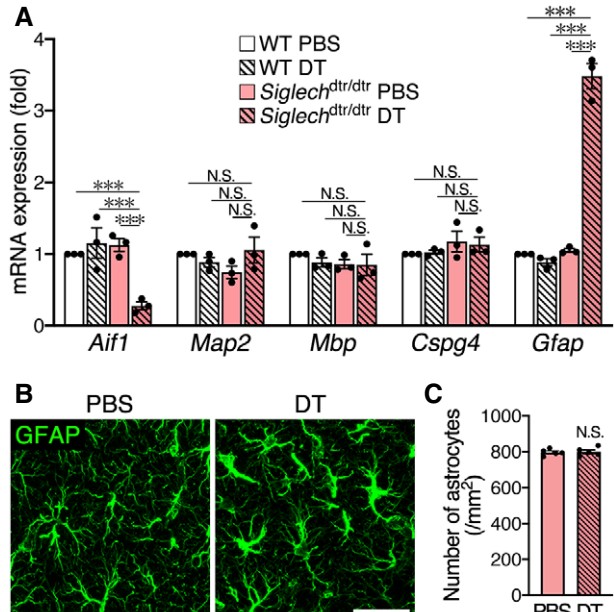

**Figure 3.  Astrocyte activation after microglial ablation.**

A   qPCR analysis of marker molecules for CNS cell types. The hippocampus of WT and *Siglech*^dtr/dtr^ mice 2 days after PBS or DT administration was analyzed (*n* = 3 animals per group, one-way ANOVA with *post hoc* Tukey's test). Results are normalized to *Gapdh* and are shown as ratios to the value of WT mice injected with PBS.

B   Immunohistochemical detection of astrocytes in *Siglech*^dtr/dtr^ mice. Hippocampal CA1 sections were prepared 2 days after PBS or DT administration, and stained with an anti-GFAP antibody. Images were acquired using the same laser power and sensitivity, and image processing was the same. Scale bar, 30 μm.

C   Astrocyte number in hippocampal CA1 of *Siglech*^dtr/dtr^ mice 2 days after PBS or DT administration (*n* = 5 animals, two-tailed unpaired Student's *t*-test).

Data information: Values show the mean ± SEM. N.S.: no significance; ***$P$ < 0.001.

genetic model (tamoxifen administration to *Cx3cr1*^CreER/+^: *Rosa26*^DTA/+^ mice; Fig EV4A, C and E) and in a pharmacological model (oral administration of PLX3397; Fig EV4B, D and F), astrocytes became activated with *Gfap* upregulation and frequently contacted microglial debris in the hippocampal CA1, demonstrating that this is a general phenomenon upon microglial ablation.

To confirm engulfment of microglial debris by astrocytes, the cytoplasm of astrocytes was stained with an S100β antibody 2 days

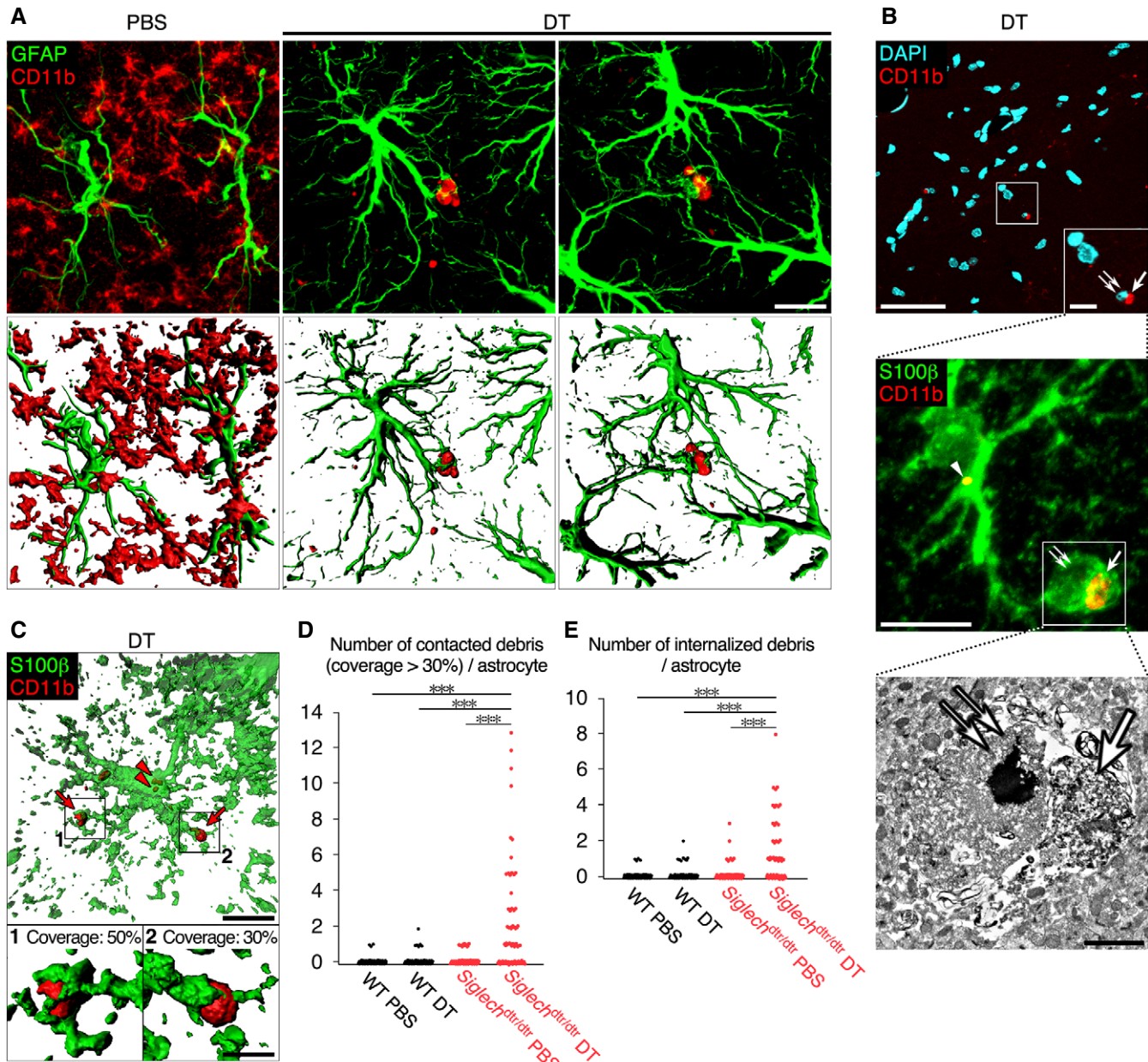

**Figure 4. Astrocytes engulf microglial debris after microglial ablation.**

A  Representative images of microglial debris surrounded by astrocyte processes in hippocampal CA1. Sections were prepared from *Siglech*dtr/dtr mice 2 days after PBS or DT administration, and stained with anti-GFAP (green) and anti-CD11b (red) antibodies. 3D images (lower row) were reconstructed from confocal images (upper row) using Imaris software. Scale bar, 10 μm.

B  CLEM analysis of a phagocytic astrocyte in hippocampal CA1 of *Siglech*dtr/dtr mice 2 days after DT administration. A phagocytic astrocyte, which engulfed degenerated microglial components, was identified by immunohistochemistry using anti-S100β (green) and anti-CD11b (red) antibodies, and DAPI (cyan) (upper and middle panels). Then, a phagocytic cup of the same cell was analyzed by electron microscopy (lower panel). Degenerated microglial cytoplasm/plasma membrane (arrow) and nucleus (double arrow) in a phagocytic cup of the astrocyte, and a piece of microglial debris internalized in astrocytic cytoplasm (arrowhead) are indicated. Scale bar, 50 μm (upper panel), 10 μm (upper panel [inset] and middle panel), and 2 μm (lower panel).

C  A representative image of a phagocytic astrocyte in hippocampal CA1 of *Siglech*dtr/dtr mice 2 days after DT administration. Sections were stained with anti-S100β (green) and anti-CD11b (red) antibodies, and confocal images were reconstructed using Imaris software. The lower two panels are higher magnification images of contacted microglial debris. Arrow: microglial debris contacted by astrocyte processes (coverage: approximately 30 or 50%). Arrowhead: microglial debris internalized in astrocyte cytoplasm. Scale bar, 10 μm (upper panel) and 2 μm (lower panels).

D, E  Quantification of the pieces of debris (CD11b+ spheres with > 0.5 μm diameter) contacted with (coverage > 30%) (D) and internalized by (E) a single astrocyte within 30-μm-thick sections (*n* = 75 cells from five animals per group). The number of debris pieces was counted in 3D images reconstructed using Imaris software. All confocal images were acquired using the same laser power and sensitivity, and image processing was the same.

Data information: ***$P < 0.01$; Mann–Whitney *U*-test.

after DT administration (Fig 4B). We stained for S100β because GFAP is an intermediate filament protein, and the entire astrocyte morphology is not visualized by GFAP immunoreactivity (Sun & Jakobs, 2012). S100β immunostaining demonstrated that astrocyte processes frequently contacted and covered microglial debris. The tip of astrocyte processes was occasionally enlarged and formed a phagocytic cup. Correlative light and electron microscopy (CLEM) demonstrated that the phagocytic cup enclosed microglial degenerated material (an arrow [cytoplasm/plasma membrane] and a double arrow [nucleus] in Fig 4B). Immunostaining of astrocyte cytoplasm also revealed that astrocytes internalized microglial debris in their cytoplasm (an arrowhead in Fig 4B). Quantitative analysis of 3D-reconstructed astrocytes indicated that a number of astrocytes, but not all, contacted and engulfed the debris in *Siglech*$^{dtr/dtr}$ mice administered DT (Fig 4C–E).

We analyzed homozygous *Siglech*$^{dtr/dtr}$ mice throughout this study because they had a higher ablation rate than heterozygotes (Appendix Fig S3A). However, as we previously reported (Konishi *et al*, 2017), microglial Siglec-H expression was considerably knocked down in homozygotes, raising the possibility that engulfment of microglial debris by astrocytes was unexpectedly caused by *Siglech* deficiency. *Gfap* mRNA levels were equivalent between WT and *Siglech*$^{dtr/dtr}$ mice in the PBS-treated group (Fig 3A), suggesting that *Siglech* deficiency was not likely to affect astrocyte activity. Nevertheless, we tested *Siglech*$^{dtr/+}$ heterozygotes, in which expression of both Siglec-H protein and *Siglech* mRNA was relatively retained compared with *Siglech*$^{dtr/dtr}$ homozygotes (Appendix Fig S3A and B). As with homozygotes, astrocytes contacted and internalized microglial debris in the hippocampal CA1 of heterozygotes (Appendix Fig S3C–F), indicating that the engulfment did not result from *Siglech* deficiency.

## Phagocytic astrocytes display a pro-inflammatory gene expression profile

To gain molecular insight into phagocytic astrocytes, we used a magnetic-activated cell sorting (MACS) system to isolate hippocampal astrocytes (Appendix Fig S4), and performed RNA-seq 2 days after administration of PBS or DT. We then performed differential expression analysis between the two groups (Table EV1–EV3). In the DT-treated group, *Gfap* was upregulated by 3.79-fold, as shown by qPCR analysis of whole hippocampus (Fig 3A), indicating the validity of this experiment. Because of the very slight contamination of microglia in our MACS isolation system (Appendix Fig S4), the expression level of microglia-enriched genes was extremely low (Appendix Fig S5). However, their downregulation was very significant because of the decreased microglial number, resulting in the inclusion of many microglia-enriched genes in the list of statistically downregulated genes (Table EV3). Therefore, gene set enrichment analysis was performed on 160 significantly upregulated genes (Table EV2), revealing enrichment of inflammation-related pathways in astrocytes after DT administration (Fig 5A). The most enriched was "tumor necrosis factor (TNF) signaling pathway". Other cytokine pathways, together with their major downstream pathway, "nuclear factor (NF)-kappa B signaling", were also significantly enriched in phagocytic astrocytes. Microglia are the major source of pro-inflammatory molecules in the CNS. However, microglial contamination was very slight in our isolation system

(Appendix Figs S4 and S5), indicating that activated astrocytes exhibited a pro-inflammatory gene expression profile.

## Astrocytes clear microglial debris predominantly using pre-expressed phagocytic receptors, Axl and Mertk

RNA-seq data of isolated astrocytes provided a molecular basis for their phagocytic activity. Gene ontology (GO) analysis of statistically upregulated genes demonstrated a statistical change in only one GO biological term "positive regulation of phagocytosis" ($P = 0.023$) among hundreds of phagocytosis- or lysosome-related terms (Table EV4), suggesting that astrocytes likely use pre-expressed phagocytic machinery for the clearance of microglial debris. Therefore, the exploration of phagocytic receptors recognizing microglial debris did not depend on upregulated genes. Among genes whose expression was detected in the RNA-seq data of control astrocytes, we focused on multiple epidermal growth factor-like domains 10 (MEGF10) and Mertk, both of which are reported as phagocytic receptors for neuronal debris and unnecessary synapses in injured and developing brain, respectively (Chung *et al*, 2013; Morizawa *et al*, 2017). In the RNA-seq data, *Megf10* was moderately and *Mertk* was highly expressed by astrocytes, both before and after DT administration (Fig 5B). Mertk, together with Tyro3 and Axl, constitutes the Tyro3/Axl/Mertk (TAM) family of phagocytic receptors (Qingxian *et al*, 2010). Tyro3, Axl, and Mertk have functional redundancy because the three family members bind to common extracellular adaptor molecules, Gas6 and protein S, both of which recognize phosphatidylserine exposed on the surface of apoptotic cells (Qingxian *et al*, 2010). In addition to *Mertk*, RNA-seq data also showed expression of *Axl* (Fig 5B). In contrast, *Tyro3* expression was very low. *Gas6* and *Pros1* (the gene encoding protein S) were both expressed by astrocytes. Although the expression of *Pros1* was slightly increased, that of other molecules was statistically unchanged after microglial ablation.

To evaluate whether MEGF10 and the TAM family act as phagocytic receptor(s) for microglial debris, we employed a co-culture system (Fig 6A). The membranes of primary microglia were labeled with a fluorescent dye, and the cells were then exposed to ultraviolet (UV) light to induce apoptosis. These cells were then co-cultured with astrocytes derived from CAG-EGFP mice. We observed that fluorescent microglial debris was internalized by astrocytes. The debris was almost entirely merged with a lysosomal marker, lysosome-associated membrane protein 1 (LAMP-1; Fig 6B), indicating that phagocytosed microglial debris was transported to astrocytic lysosomes to be digested. Flow cytometric analysis detected fluorescence of internalized microglial debris in GLAST$^+$ astrocytes (Fig 6C).

In line with the RNA-seq data (Fig 5B), MEGF10, as well as Axl and Mertk but not Tyro3, was detected in primary cultured astrocytes by Western blotting (Figs 6D and EV5A). First, we knocked down MEGF10 expression and evaluated its effect on uptake of microglial debris by flow cytometric quantification. MEGF10 expression was successfully knocked down to almost no detectable level by two siRNAs (Fig EV5A and B); however, MEGF10 knockdown had no effect on phagocytosis of microglial debris by astrocytes (Fig EV5C). We then tested TAM family members. We knocked down expression of all three members using two siRNAs for each gene (Fig 6D and E). The knockdown of Axl and Mertk was

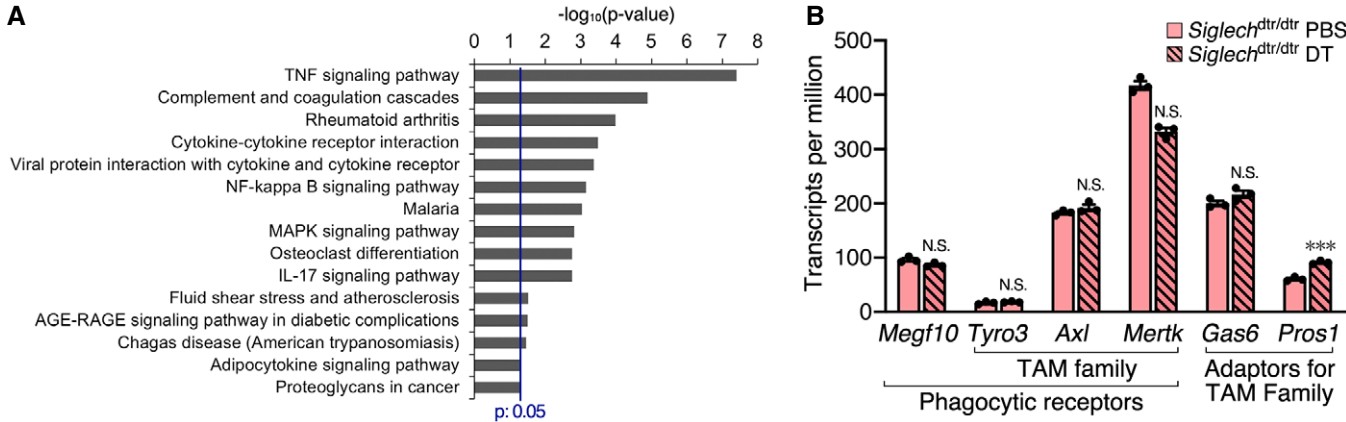

**Figure 5. Gene expression analysis of isolated hippocampal astrocytes by RNA-seq before and after microglial ablation.**

A  Gene sets enriched in astrocytes 2 days after microglial ablation, demonstrated using g:Profiler with the KEGG database.

B  Expression level of phagocytic receptors and their extracellular adaptor molecules ($n$ = 3 independent experiments). Values show the mean ± SEM. N.S.: no significance; ***$q$ < 0.001. The probability values for differential expression analysis were computed with edgeR followed by a correction of multiple testing with $q$-value.

evaluated by Western blotting. However, as described above, Tyro3 expression was undetectable by Western blotting, and *Tyro3* knockdown was confirmed by quantitative real-time PCR (qPCR). Flow cytometric quantification showed that, like control siRNA, *Tyro3* siRNAs did not affect uptake of microglial debris (Fig 6F), which was expected because almost no Tyro3 protein was detected in control astrocytes (Fig 6D). By contrast, knockdown of *Axl* or *Mertk* suppressed the engulfment (Fig 6F), although the suppressive effects were small. Given the functional redundancy of TAM family members, we then examined the combination of *Axl* and *Mertk* siRNAs, which produced a ~50% reduction in debris uptake (Fig 6G). These results indicate that functional phagocytic receptors, Axl and Mertk, cooperatively act as the main phagocytic receptors for microglial debris, although additional receptors may participate.

**Phagocytic activity of astrocytes compensates for microglial dysfunction in *Irf8*$^{-/-}$ mice**

Microglial ablation models reveal clearance of microglial debris by astrocytes, suggesting that astrocytic phagocytosis compensates for microglial dysfunction. However, this phenomenon was observed in artificial conditions, in which most microglia disappeared in a short period of time. Therefore, we tested whether astrocytes perform phagocytosis in the presence of microglia whose phagocytic activity was suppressed (Fig 7A). We selected mutant mice, in which some microglial functions are impaired: mice deficient for *Cx3cr1* (the gene encoding the fractalkine receptor, which regulates microglial activities) (Zhan *et al*, 2014), *Csf1* (the gene encoding a cytokine essential for microglial differentiation and viability) (Nandi *et al*, 2012), *Tyrobp* (the gene encoding DNAX-activating protein of 12 kDa [DAP12], a signal transduction molecule of microglial activation receptor, triggering receptor expressed on myeloid cells 2 [TREM2]) (Kobayashi *et al*, 2016), and *Irf8* (the gene encoding a transcription factor essential for microglial differentiation) (Masuda *et al*, 2012). Spontaneous apoptosis occurs in the developing brain (Oppenheim, 1991). The neonatal

brain contains many spontaneous apoptotic cells; however, it was not suitable for this study because glial cells were immature. Our preliminary immunohistochemical experiment indicated that the cerebral cortex of 3-week (3-w)-old mice contained a small but consistent number of cleaved caspase-3$^+$ apoptotic cells. Prior to testing for a decline in microglial phagocytic activity, we quantified microglial density in the cerebral cortex of each mutant strain by Iba1 immunostaining (Fig 7B). Iba1 immunoreactivity was substantially attenuated in *Irf8*$^{-/-}$ microglia (Appendix Fig S6), probably because *Aif1* (the gene encoding Iba1) is a target of interferon regulatory factor 8 (IRF8) (Kurotaki *et al*, 2018); therefore, as an alternative, we performed CD11b immunostaining on *Irf8*$^{-/-}$ mice. This clearly demonstrated that all strains showed the same microglial density in the cerebral cortex (Fig 7C). Using these mutant mice, we evaluated whether astrocytes were able to recognize and engulf the debris of apoptotic cells.

All apoptotic cells were clearly encircled by microglial processes in WT mice (Fig 7D and E), indicating that microglia are the primary phagocytes for cellular debris in the normal brain, as is well established. We then tested whether astrocytes performed phagocytosis upon decline of microglial phagocytic activity. The number of spontaneous apoptotic cells was statistically comparable among all strains of mice (Appendix Fig S7). In *Irf8*-deficient mice, compared with the other mutant strains, microglia exhibited a hypertrophic shape with fewer, shorter, and less-branched processes as reported in previous studies (Hagemeyer *et al*, 2016), and the percentage of apoptotic cells surrounded by microglia was considerably decreased to almost half (Fig. 7D and E). Instead, astrocyte processes frequently surrounded apoptotic cells, occasionally forming a phagocytic cup (Fig 7D). The rate of apoptotic cells surrounded by astrocyte processes in *Irf8*$^{-/-}$ mice was significantly higher than that in WT and other mutant mice (Fig 7F). For apoptotic cells not encircled by microglial processes, the percentage of cells surrounding by astrocytes was 83.3 ± 9.1% (mean ± SEM, $n$ = 5) in *Irf8*$^{-/-}$ mice. These results indicate that astrocytes act as a substitute for phagocytosis-impaired microglia.

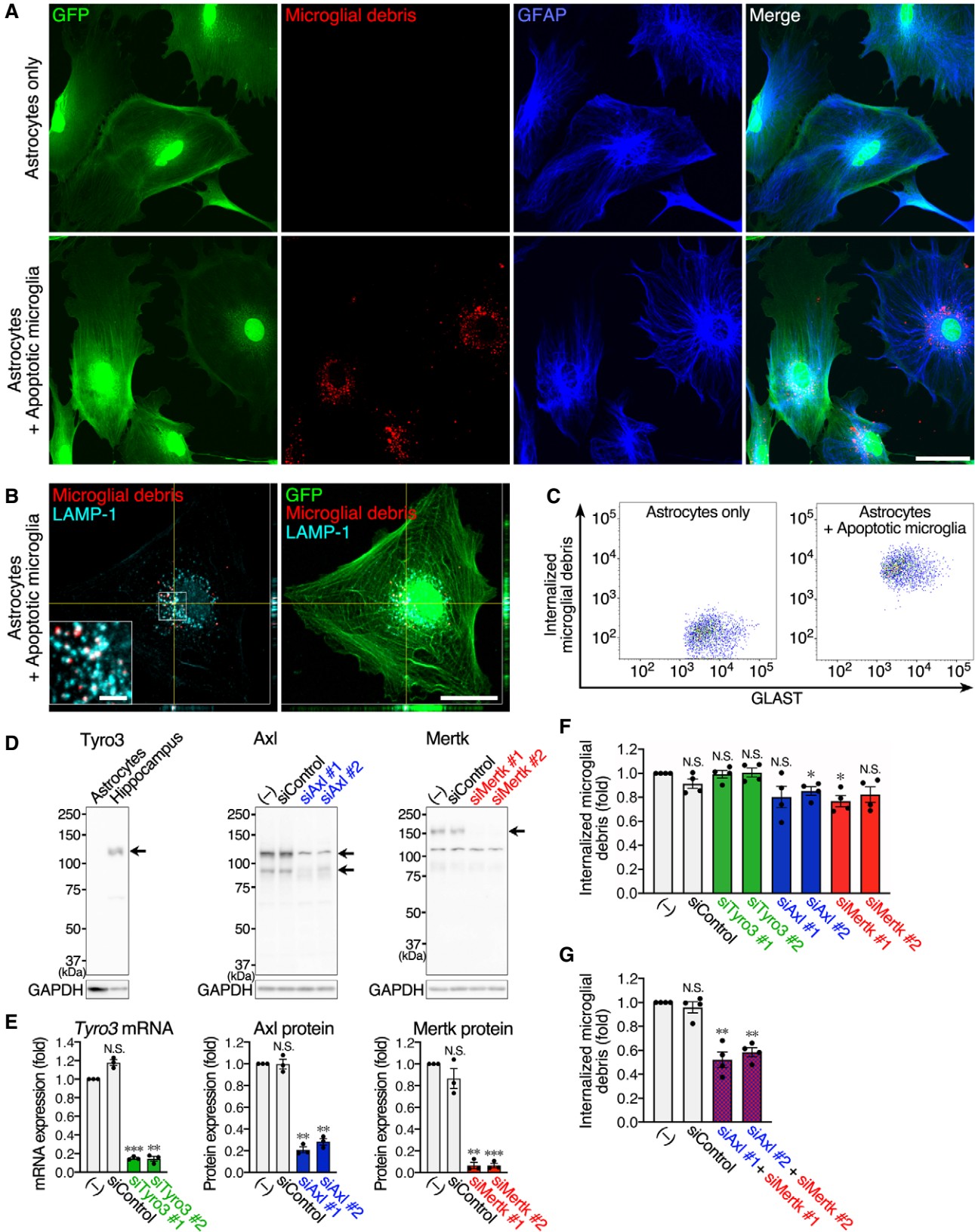

**Figure 6.**

◄ **Figure 6. Axl and Mertk in cultured astrocytes serve as phagocytic receptors for microglial debris.**

A, B   Cultured astrocytes engulf microglial debris. CAG-EGFP mouse-derived primary astrocytes were co-cultured with or without fluorescence-labeled apoptotic microglia. (A) Fluorescence images of GFP (green), microglial debris (red), and GFAP immunoreactivity (blue). Images were acquired using the same laser power and sensitivity, and image processing was the same in the two images (red). Scale bar, 50 μm. (B) Fluorescence images of GFP (green), microglial debris (red), and LAMP-1 immunoreactivity (cyan). Scale bar, 30 μm and 5 μm (inset).

C   Flow cytometric analysis of uptake of microglial debris by cultured astrocytes. Fluorescence of internalized microglial debris was measured in GLAST$^+$ astrocytes co-cultured with or without fluorescence-labeled apoptotic microglia.

D   Expression of TAM family members and their knockdown by siRNA in cultured astrocytes. Tyro3, Axl, and Metk in cultured astrocytes or hippocampal tissue were examined by Western blotting. Arrows indicate the corresponding protein band(s). The secondary antibody against goat IgG generates a ~115 kDa non-specific band. The ~115 kDa band in the Axl image is the Axl-specific band overlapping the non-specific band. The ~115 kDa band in the Mertk image is the non-specific band. Axl and Mertk expression was knocked down by two siRNAs. GAPDH expression is shown as an internal control.

E   Quantification of siRNA knockdown in cultured astrocytes ($n = 3$ independent experiments, one-way ANOVA with repeated measures). Knockdown of *Tyro3* mRNA was examined by qPCR because of the very low expression level even in control astrocytes. Knockdown efficiency of Axl and Mertk was quantified from Western blots. Results are normalized to *Gapdh* or GAPDH, and are shown as ratios to the value of non-transfected astrocytes.

F, G   Flow cytometric analysis of microglial debris uptake by cultured astrocytes. After knockdown of TAM family members, astrocytes were co-cultured with fluorescence-labeled apoptotic microglia. Fluorescence of internalized microglial debris in GLAST$^+$ astrocytes was quantified by flow cytometry ($n = 4$ independent experiments, one-way ANOVA with repeated measures). Results are shown as ratios to the value of non-transfected astrocytes.

Data information: Values show the mean ± SEM. N.S.: no significance; *$P < 0.05$, **$P < 0.01$, ***$P < 0.001$.
Source data are available online for this figure.

## Discussion

The present study revealed that astrocytes possess phagocytic machinery, which can be actuated in the event of microglial dysfunction. In *Siglech*$^{dtr}$ mice, microglia-specific ablation produced a significant amount of microglial debris immediately after DT injection, but the debris was completely cleared in few days (Fig 1). We found that activated astrocytes, not non-microglial mononuclear cells, engulfed microglial debris using normally expressed phagocytic receptors, Axl and Mertk (Fig 2–6). This phenomenon was also observed in non-ablation models, such as *Irf8*$^{−/−}$ mice (Fig 7); the impaired microglial phagocytosis resulting from *Irf8* deficiency was compensated by astrocytes.

We observed clearance of microglial debris in hippocampal CA1 during the 2–4 days post-DT injection (Fig 1A and D). In this system, a small population of microglia were able to escape death (Fig 1A and C). However, the survived microglia exhibited abnormal shapes (Appendix Fig S1), probably from DT damage, and did not interact with microglial debris (Fig 1E), indicating the existence of a microglia-independent debris clearance system. Elmore *et al* (2014) first reported PLX compounds to be effective microglial depletion reagents, and they described the possibility of microglia-independent clearance of microglial debris. However, neither they nor others reporting microglial ablation models addressed the mechanisms underlying the clearance of microglial debris (Parkhurst *et al*, 2013; Bruttger *et al*, 2015; Waisman *et al*, 2015; Cronk *et al*, 2018; Han *et al*, 2019), and the possibility remained that the microglial debris was cleared by other CNS-resident macrophages and/or infiltrating monocytes. The present study ruled out those possibilities by using *Siglech*$^{dtr}$ mice. As expected by the specificity of Siglec-H expression (Fig EV1), the two candidate non-microglial mononuclear lineage populations, CCR2$^+$Ly6C$^{high}$ inflammatory monocytes and CD206$^+$ perivascular macrophages, were alive even after DT administration (Fig 2B and D). However, CCR2$^+$ monocytes had not infiltrated hippocampal parenchyma 2–4 days after DT administration (Fig 2A), which was also indicated by the absence of GFP$^+$ blood-derived cells in hippocampal parenchyma of parabionts in the recovery phase, 7 days post-DT administration (Fig EV2C). In addition, perivascular macrophages remained localized in perivascular regions (Fig 2C).

Therefore, we concluded that neither cell type acted as scavengers for microglial debris. In contrast to phagocytic activity of infiltrated monocytes, which are known to significantly affect neuronal fates under inflamed or injured conditions (Yamasaki *et al*, 2014; Ritzel *et al*, 2015), that of perivascular macrophages has received less attention. Although perivascular macrophages phagocytose substances within perivascular spaces (Kida *et al*, 1993), our results indicate that they do not respond to cellular debris in CNS parenchyma, which is insulated from perivascular spaces by astrocytic endfeet (Faraco *et al*, 2017).

Astrocytes, which are not professional phagocytes, became activated after microglial ablation (Fig 3). Astrocyte activation is a common phenomenon in different kinds of previously reported microglial ablation models (Elmore *et al*, 2014; Bruttger *et al*, 2015; Peng *et al*, 2016; Lund *et al*, 2018; Rubino *et al*, 2018); however, the functional consequences of this activation are unclear. Our histological data demonstrated that the activated astrocytes not only contacted but also engulfed microglial debris using their processes (Figs 4 and EV3). RNA-seq analysis demonstrated that pro-inflammatory pathways were enriched in the activated astrocytes (Fig 5A). What then is the significance of pro-inflammatory activation of astrocytes? Bruttger *et al* (2015) reported that microglial ablation causes a parenchymal cytokine storm. Considering our results, and also suggested by Bruttger *et al*, the primary source of the cytokine storm is likely to be activated astrocytes. Bruttger *et al* also demonstrated pro-inflammatory cytokine IL-1 as a stimulator of microglial proliferation for repopulation. Taken together, the pro-inflammatory *milieu* produced by activated astrocytes may contribute to the recovery of microglial number.

Phagocytic activity of astrocytes has been reported, although far less is known about astrocyte phagocytosis compared with microglial phagocytosis. Astrocytes engulf cellular components, such as synapses, axons, and myelin *in vivo* (Nguyen *et al*, 2011; Chung *et al*, 2013; Ponath *et al*, 2017) and dying or dead cells *in vitro* and *in vivo* (Iram *et al*, 2016; Morizawa *et al*, 2017). Morizawa *et al* (2017) recently demonstrated that astrocytes actively engulf neuronal debris in ischemic penumbra region and, in line with our results, they showed Mertk to be a phagocytic receptor on astrocytes for neuronal debris, although they did not examine a synergistic function of Mertk with another family member, Axl (Fig 6).

Although the clearance of dying or dead cells by astrocytes has been shown, its functional significance in comparison with microglial phagocytosis remains obscure. As previously established, we confirmed microglia, not astrocytes, as *bona fide* primary phagocytes for spontaneous apoptotic cells in the normal brain (Fig 7). We then showed that in *Irf8⁻/⁻* mice, with phagocytosis-impaired microglia, astrocytes elicit their phagocytic activity to compensate for declined microglial function (Fig. 7). Abiega *et al* (2016) suggested this possibility in the hippocampus of kainic acid-induced seizure model, in which microglial phagocytic activity was impaired. They

observed engulfment of apoptotic cells by astrocytes as well as by microglia after seizure and therefore proposed a compensatory function of astrocytes. In addition to epileptic injury, dysfunction of microglial phagocytosis is suggested to be associated with neurodegenerative diseases such as Alzheimer's disease (Krabbe *et al*, 2013; Lucin *et al*, 2013; Gabande-Rodriguez *et al*, 2020). It is also generally thought that microglial phagocytosis deteriorates with aging, causing a detrimental impact on CNS homeostasis (Gabande-Rodriguez *et al*, 2020; Pluvinage *et al*, 2019). In injured, neurodegenerative, or aged conditions, a compensatory function of

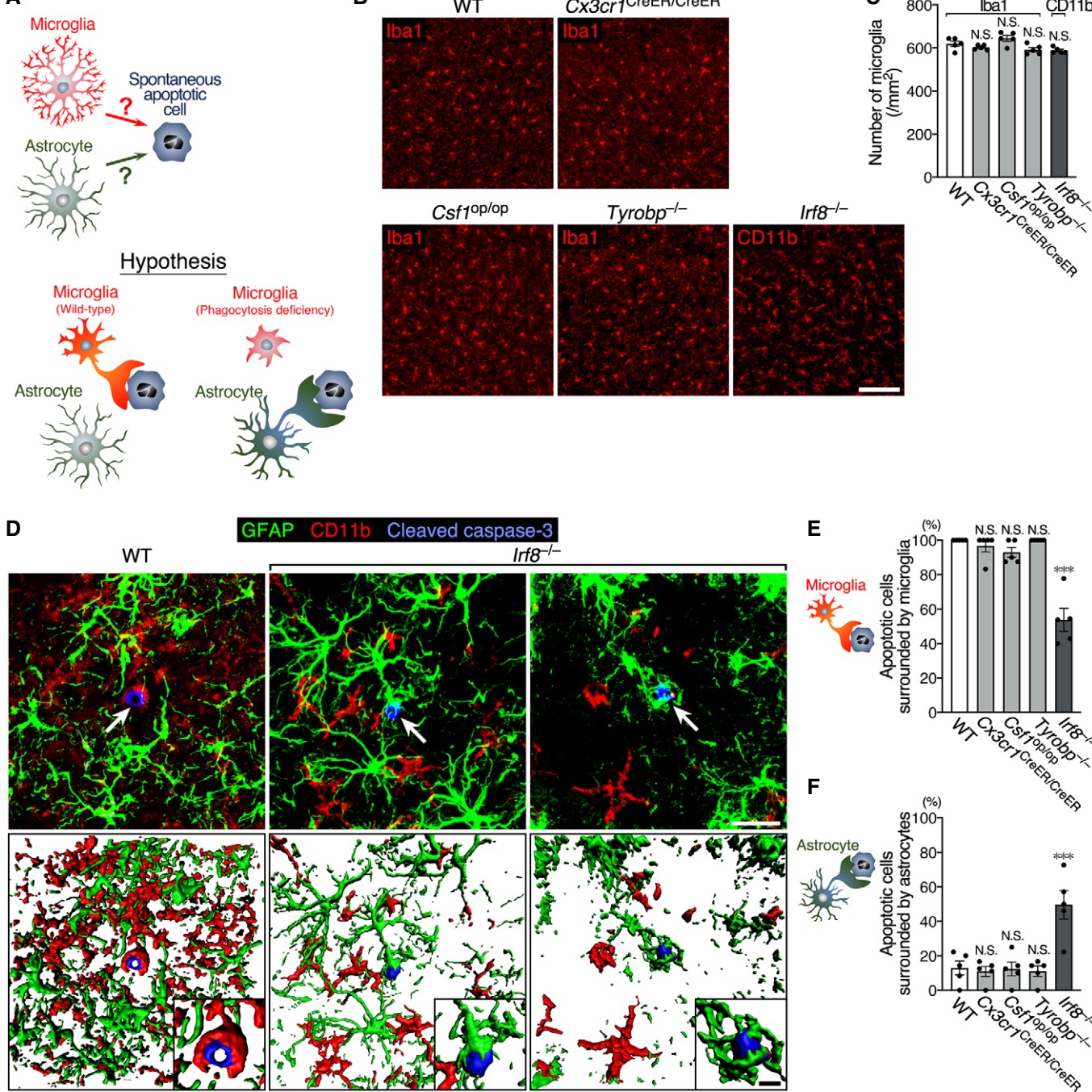

**Figure 7.**

◄  **Figure 7. Astrocytes elicit their phagocytic activity upon microglial dysfunction.**

A   A schematic drawing of the hypothesis that astrocytes perform phagocytosis on behalf of phagocytosis-deficient microglia.

B   Representative images of microglial distribution in the cerebral cortex of WT and mutant mice at 3 w. Sections were stained with anti-Iba1 or anti-CD11b antibodies (red). Scale bar, 100 μm.

C   Quantification of microglial number (*n* = 5 animals per group, one-way ANOVA). Note that *Irf8*$^{-/-}$ mice were analyzed by CD11b immunostaining, while other strains were analyzed by Iba1 staining.

D   Representative images of a spontaneous apoptotic cell and surrounding glial cells in the cerebral cortex of WT and *Irf8*$^{-/-}$ mice at 3 w. Sections were stained with anti-GFAP (green), anti-CD11b (red), and anti-cleaved caspase-3 (blue) antibodies. 3D images (lower row) were reconstructed from confocal images (upper row) using Imaris software. Arrows indicate cleaved caspase-3$^+$ apoptotic cells. Insets are higher magnification images around apoptotic cells. Scale bar, 20 μm and 5 μm (insets).

E   Percentage of spontaneous apoptotic cells surrounded by microglia in WT and mutant mice. 3D images reconstructed from confocal images were analyzed (*n* = 36–50 cells from five animals per group, Kruskal–Wallis test with *post hoc* Dunn's test).

F   Percentage of spontaneous apoptotic cells surrounded by astrocyte processes in WT and mutant mice (*n* = 36–50 cells from five animals per group, one-way ANOVA with *post hoc* Tukey's test).

Data information: Values show the mean ± SEM. N.S.: no significance; ***$P$ < 0.001.

astrocytes may contribute to maintaining debris clearance. Further studies, using other mouse models of injury, neurodegeneration or aging, and examination of human brains will further uncover the functional significance of astrocytic phagocytosis.

We show that phagocytic activity of astrocytes is triggered in response to microglial dysfunction. RNA-seq data indicate that astrocytes normally possess phagocytic machinery and stand by for engulfment of apoptotic cells, raising the question what induces their phagocytic action. A plausible inducer is the remaining debris, which cannot be cleared by microglia. Microglia are known to rapidly respond to dying cells for phagocytic clearance (Koizumi *et al*, 2007). Indeed, microglia acted as primary phagocytes for spontaneous apoptotic cells in WT mice (Fig 7). However, insufficient clearance of dead cells, which results from microglial dysfunction or over-capacity, as discussed above, possibly induces the phagocytic action of astrocytes. Alternatively, dysfunctional or a decreased number of microglia may also be an inducer. Recent studies show that microglial activity significantly affects astrocyte phenotype (Liddelow *et al*, 2017; Shinozaki *et al*, 2017). If microglia inhibit the phagocytic activity of astrocytes in a soluble factor- or contact-dependent manner in the healthy CNS, the phagocytic activity of astrocytes may emerge upon a decrease in the number or activity of microglia. It also remains unknown whether astrocytes act as spare phagocytes for live cellular components such as synapses and axons (Nguyen *et al*, 2011; Chung *et al*, 2013). Further studies are needed to resolve these important questions.

In conclusion, rapid removal of dying or dead cells is crucial for the maintenance of tissue homeostasis and for the inhibition of excessive inflammatory responses. Failure to clear cell debris causes undesired inflammatory responses, which can eventually damage surrounding tissues or lead to autoimmune diseases (Arandjelovic & Ravichandran, 2015). In the CNS, astrocytes with phagocytic receptors may stand by in case microglia are unable to remove cell debris. Investigation of further phagocytic properties of astrocytes may lead to new therapies that accelerate efficient debris clearance from the aged, injured, or degenerated CNS.

# Materials and Methods

## Animals

WT mice were purchased from Charles River Laboratories Japan, and CAG-EGFP mice were purchased from SLC Japan. *Ccr2*$^{RFP}$

(stock number: 017586), *Rosa26*$^{DTA}$ (stock number: 009669), and *Csf1*$^{op}$ (stock number: 00231) mice were obtained from the Jackson Laboratory. *Siglech*$^{dtr}$ (B6.Cg-*Siglech*$^{<tm1.1Ksat>}$ mice; deposited in RIKEN BioResource Center [accession number: RBRC05658]) (Takagi *et al*, 2011), *Cx3cr1*$^{CreER}$ (Yona *et al*, 2013), *Tyrobp*$^{-/-}$ (*Dap12*$^{-/-}$) (Kaifu *et al*, 2003), and *Irf8*$^{-/-}$ (Holtschke *et al*, 1996) mice were described previously. All mice had a C57BL/6 background, except for *Csf1*$^{op}$ (B6C3Fe background). Male mice aged 3 w or 8–16 w (adult mice) were used for *in vivo* analyses. Male or female mice at postnatal day (P) 0–2 were used to prepare microglia and astrocytes for primary culture. All studies were approved by the local animal ethics committee of Nagoya University (approval numbers: 27204, 28303, 29281, 30178, and 31072). All experimental procedures were conducted in accordance with the guidelines for animal experiments of Nagoya University Graduate School of Medicine, the Animal Protection and Management Law of Japan (No. 105), and the 3R principle (replacement, refinement, and reduction).

## Microglial ablation

According to our previous study (Konishi *et al*, 2017), DT (50 μg/kg) dissolved in PBS (Sigma-Aldrich) or the same amount of PBS was intraperitoneally injected into adult *Siglech*$^{dtr/+}$ or *Siglech*$^{dtr/dtr}$ mice. Brains were analyzed at 2, 4, 7, and 28 days after administration.

*Cx3cr1*$^{CreER/+}$:*Rosa26*$^{DTA/+}$ mice were subcutaneously administered tamoxifen (200 mg/kg; Sigma-Aldrich) dissolved in corn oil every 24 h for 5 days. Three days after the final administration (7 days after the first administration), mice were subjected to immunohistochemistry or qPCR.

PLX3397 (ChemScene; 2.5 mg) (Elmore *et al*, 2014) was suspended in 200 μl 0.5% methyl cellulose 400 and orally administrated to WT mice every 12 h (at 9 a.m. and 9 p.m.) for 3 days. Twelve hours after the final administration, mice were subjected to immunohistochemistry or qPCR.

## Immunohistochemistry

Immunohistochemistry was performed according to our previous study with slight modification (Kobayashi *et al*, 2016). Briefly, mice were transcardially perfused with Zamboni's fixative (0.1 M phosphate buffer [PB] containing 2% paraformaldehyde [PFA] and 0.2%

picric acid). Brains, spinal cords, and inguinal lymph nodes were dissected, dehydrated in 25% sucrose in 0.1 M phosphate buffer, and then frozen in dry ice. Floating sections were cut on a microtome at 30 μm and reacted with primary antibodies diluted in a blocking buffer (0.01 M PBS containing 1% bovine serum albumin [BSA]), 0.1% Triton X-100, and 0.1% $NaN_3$). The following primary antibodies were used: rabbit anti-Iba1 (WAKO #019-19741, RRID: AB_839504), goat anti-Iba1 (Abcam #ab5076, RRID: AB_2224402), rabbit anti-cleaved caspase-3 (Cell Signaling Technology #9664, RRID: AB_2070042), rabbit anti-Ki-67 (Thermo Fisher Scientific #RM-9106, AB_2335745), goat anti-CD206 (R&D systems #AF2535, RRID: AB_2063012), rat anti-CD31 (BD Biosciences #550274, RRID: AB_393571), goat anti-GFAP (Abcam #ab53554, RRID: AB_880202), rat anti-CD11b (Bio-Rad #MCA711, RRID: AB_321292), chicken anti-S100β (Synaptic Systems #287006, RRID: AB_2713986), rabbit anti-S100β (Abcam #ab52642, RRID: AB_882426), and sheep anti-Siglec-H (Zhang *et al*, 2006). After reaction with secondary antibodies conjugated with Alexa Fluor 488, 594, or 647 (Thermo Fisher Scientific or Abcam), and 4′,6-diamidino-2-phenylindole (DAPI) staining, sections were mounted on slide glasses. Images were taken using a confocal FV10i microscope (Olympus). For 3D image analyses, confocal images were reconstructed using Imaris software (version 8.1.2, Zeiss).

## Quantitative histological analyses

Numbers of live microglia, apoptotic microglia, Ki-67$^+$ microglia, and pieces of microglial debris in hippocampal CA1 after microglial ablation. Coronal sections (30 μm thick) of adult Siglech$^{dtr/dtr}$ mice were prepared at 0 (no injection), 2, 4, 7, and 28 d after DT administration and were stained with anti-Iba1 and anti-Ki-67 antibodies and DAPI. Images of the hippocampal CA1 region were taken at ×60 magnification. The number of live microglia with a normal nucleus, apoptotic microglia with a pyknotic or fragmented nucleus, Ki-67$^+$ microglia, and pieces of microglial debris (Iba1$^+$ spheres with a diameter > 2 μm and no nuclei) were counted and normalized to the area. A total of 40 images (1 field/section, 8 sections/animal, 5 animals) were analyzed for each time point by investigators blind to the animals' grouping.

Morphometric alteration of microglia in hippocampal CA1 after microglial ablation. Coronal sections (30 μm thick) of adult Siglech$^{dtr/dtr}$ mice were prepared at 0 (no injection), 2, 4, 7, and 28 d after DT administration and were stained with an anti-Iba1 antibody and DAPI. Images of the hippocampal CA1 region were taken at ×60 magnification. Microglia with nuclei were randomly chosen from the confocal images, and their morphologies were quantified using the skeleton analysis plug-in tool in Image J as previously described (Young & Morrison, 2018). A total of 50 microglia (10 cells/animal, 5 animals) in each group were analyzed by investigators blind to the animals' grouping.

Numbers of perivascular macrophages and astrocytes in hippocampal CA1 after microglial ablation. Coronal sections (30 μm thick) of adult Siglech$^{dtr/dtr}$ mice were prepared 2 days after administration of PBS or DT, and stained with anti-CD206 or anti-GFAP antibody. Images of the hippocampus were taken at ×18 magnification. The number of CD206$^+$ perivascular macrophages or GFAP$^+$ astrocytes within the CA1 region were counted and normalized to the area. A total of 20 images (1 field/section, 4

sections/animal, 5 animals; for perivascular macrophages) or 15 images (1 field/section, 3 sections/animal, 5 animals; for astrocytes) were analyzed for each group by investigators blind to the animals' grouping.

Phagocytosis of microglial debris by astrocytes in hippocampal CA1. Coronal sections (30 μm thick) of adult WT, Siglech$^{dtr/+}$, or Siglech$^{dtr/dtr}$ mice were prepared 2 d after administration of PBS or DT and were double-stained with anti-CD11b and anti-S100β antibodies. Images of a randomly selected single S100β$^+$ astrocyte, whose cell body was entirely present within a 30-μm-thick section, were taken at ×240 magnification. After 3D reconstruction of cell morphology by cell surface rendering using Imaris software, the number of CD11b$^+$ spheres (diameter > 0.5 μm), which were more than 30% covered by astrocyte processes or internalized in astrocytes, was counted for the single astrocyte. In the acquired images, cell bodies or processes of other astrocytes were present around the chosen astrocyte; however, they were excluded from the analysis. A total of 75 astrocytes (5 cells/section, 3 sections/animal, 5 animals) in each group were analyzed by investigators blind to the animals' grouping.

Number of microglia in the cerebral cortex of 3-w-old mice. Hippocampal-level coronal sections (30 μm thick) were prepared from 3-w-old WT, Cx3cr1$^{CreER/CreER}$, Csf1$^{op/op}$, Tyrobp$^{-/-}$, and Irf8$^{-/-}$ mice and stained with anti-Iba1. In Irf8$^{-/-}$ mice, microglia were stained by the anti-CD11b antibody and not by the anti-Iba1 antibody because of the attenuated expression of Iba1. Images of the ventral part of the cerebral cortex were taken at ×10 magnification. The number of Iba1$^+$ or CD11b$^+$ microglia was counted and normalized to the area. A total of 15 images (1 field/section, 3 sections/animal, 5 animals) in each group were analyzed by investigators blind to the animals' grouping.

Interaction of spontaneous apoptotic cells with microglia and astrocytes in the cerebral cortex of 3-w-old mice. Hippocampal-level coronal sections (30 μm thick) were prepared from 3-w-old WT, Cx3cr1$^{CreER/CreER}$, Csf1$^{op/op}$, Tyrobp$^{-/-}$, and Irf8$^{-/-}$ mice, and triple-stained with anti-cleaved caspase-3, anti-CD11b, and anti-GFAP antibodies. Images of cleaved caspase-3$^+$ apoptotic cells in the ventral part of the cerebral cortex were taken at ×120 magnification (36–50 cells in total, at least 5 cells/animal, 5 animals). The percentage of apoptotic cells surrounded by CD11b$^+$ microglial cell bodies or processes was determined in WT, Cx3cr1$^{CreER/CreER}$, Csf1$^{op/op}$, Tyrobp$^{-/-}$, and Irf8$^{-/-}$ mice. Apoptotic microglia, which were double-positive for cleaved caspase-3 and CD11b, were excluded in this analysis because when CD11b was expressed in both apoptotic and healthy surrounding microglia, evaluation of their interaction was difficult. After 3D reconstitution of images by Imaris software, the interaction of apoptotic cells with astrocyte processes was evaluated in all strains of mice. Apoptotic astrocytes, which were double-positive for cleaved caspase-3 and GFAP, were not analyzed for the same reason that apoptotic microglia were not analyzed, which is described above. Data analysis was blind.

## Parabiotic surgery

The surgical procedure was performed as described in our previous paper with slight modification (Tashima *et al*, 2016). Age-matched adult Siglech$^{dtr/dtr}$ and CAG-EGFP mice were housed in the same cage for at least 1 w before surgery. Mirror-imaged incisions from

elbow to knee were made in each pair under 2% isoflurane anesthesia after shaving the incision site. The tendons of the triceps brachii and quadriceps femoris of each pair were sutured, and the skin of each pair was stapled together. Blood chimerism was determined by flow cytometry 3 w after surgery ($n = 5$). The parabionts were then administrated with DT for microglial ablation.

### Flow cytometry

Flow cytometry was performed using FACSCanto II (BD Biosciences) or FACSVerse, and data were analyzed by FlowJo (Tree Star) or FACSDiva software (BD Biosciences).

For chimerism analysis of parabionts, peripheral blood of WT mice was obtained from a tail vein 3 w after surgery ($n = 5$). After removal of red blood cells using lysis solution (BD Pharm Lyse, BD Biosciences), the percentage of GFP$^+$ blood cells was determined.

To analyze inflammatory monocytes in peripheral blood after microglial ablation, prior to transcardiac perfusion of fixative, venous blood was withdrawn from the right ventricle of Siglech$^{dtr/dtr}$:Ccr2$^{RFP/+}$ mice, which had been administrated PBS or DT 2 d previously. After removal of red blood cells, Fc receptors were blocked with anti-CD16/CD32 antibodies (Thermo Fisher Scientific #14-0161-81, RRID: AB_467132), and cells were stained with an anti-Ly6C-APC antibody (BioLegend #108411, RRID: AB_313376). The percentage of RFP$^+$Ly6C$^{high}$ inflammatory monocytes was determined.

Astrocytic phagocytosis of microglial debris was quantified in vitro using cultured astrocytes that were co-cultured with PKH67 (Sigma-Aldrich)-labeled apoptotic microglia for 10–12 h. Cells were detached from the culture dish by the treatment of 0.025% trypsin (Thermo Fisher Scientific) at 37°C for 3 min, fixed with 2% PFA at 4°C at least for 5 min, and washed in PBS containing 1% BSA. After permeabilization with 0.1% Triton X-100, Fc receptors were blocked with anti-CD16/CD32 antibodies. Then, cells were stained with an anti-GLAST-APC antibody (Miltenyi Biotec #130-098-803, RRID: AB_2660783), and the PKH67 signal in GLAST$^+$ astrocytes was analyzed.

To confirm the enrichment of astrocytes in the ACSA-2$^+$ fraction of the MACS isolation system, cells in CD11b$^+$, ACSA-2$^+$, and the flow-through fractions were fixed and permeabilized using the Fixation/Permeabilization Solution Kit (BD Biosciences). After staining with anti-CD45-PE-Cy7 (BD Biosciences #552848, RRID: AB_394489) and anti-GFAP-PE (Millipore #FCMAB257P, RRID: AB_10807277) antibodies, the percentages of astrocytes and microglia in the ACSA-2$^+$ fraction were determined. The percentage of GFAP$^+$ cells in the ACSA-2$^+$ fraction may be underscored because a minor population of hippocampal astrocytes are GFAP$^-$ (Jinno, 2011).

### EAE induction

Adult Siglech$^{dtr/dtr}$:Ccr2$^{RFP/+}$ mice were immunized subcutaneously with 200 μg MOG$_{35-55}$ peptide (MBL Life Science) in complete Freund's adjuvant emulsion, followed by intraperitoneal injection of 200 ng pertussis toxin (Wako; Konishi et al, 2017). Upon the appearance of hindlimb paralysis, the ventral lumbar spinal cord was analyzed by immunohistochemistry.

### qPCR

Hippocampi, cerebral cortex, thalamus, and medulla collected from adult WT, Siglech$^{dtr/+}$, Siglech$^{dtr/dtr}$, and Cx3cr1$^{CreER/+}$:Rosa26$^{DTA/+}$ mice ($n = 3$), or cultured primary astrocytes ($n = 3$) were analyzed by qPCR. mRNA was purified using the acid guanidine iso-thiocyanate/phenol/chloroform extraction method and converted to cDNA by reverse transcriptase SuperScript III according to the manufacturer's protocol (Thermo Fisher Scientific). Gene expression was analyzed using Fast SYBR Green Master Mix (Applied Biosystems) and the StepOnePlus system (Applied Biosystems). Cycling parameters were as follows: 1 cycle of 95°C for 20 s, 40 cycles of 95°C for 3 s, and 60°C for 30 s. Primer sets were as follows: Gapdh (sense 5′-TGACGTGCCGCCTGGAGAAA-3′, antisense 5′-AGTGTAGCCCAAGA TGCCCTTCAG-3′), Aif1 (gene encoding Iba1) (sense 5′-GGAT CTGCCGTCCAAAC-3′, antisense 5′-GCATTCGCTTCAAGGACA-3′), Map2 (sense 5′-TCCATCTCTTCAGCACGAC-3′, antisense 5′-TAG TAGGTGTGGAGGTGCCA-3′), Mbp (sense 5′-GCTCCCTGCCCCAGA AGT-3′, antisense 5′-TGTCACAATGTTCTTGAAGAAATGG-3′), Cspg4 (gene encoding NG2) (sense 5′-GTGGCCTTCACGATCA-3′, antisense 5′-CGATGGTGTAGACCAAGT-3′), Gfap (sense 5′-TGAATCGCTGGAG GAGGAGA-3′, antisense 5′-TTGGCCACATCCATCTCCAC-3′), Siglech (sense 5′-GGAGAGACCAGCAACACACA-3′, antisense 5′-TCCAGT TGGCACCATCATCC-3′), and Tyro3 (sense 5′-AGAGTTTGGATCAG TGCGGG-3′, antisense 5′-ACTCCTTCATGCAAGCTGCT-3′). Melt curve analysis was performed to confirm amplicon specificity. Data were normalized with Gapdh, and fold changes were determined using the $2^{-\Delta\Delta Ct}$ method.

### CLEM

Two days after DT administration, Siglech$^{dtr/dtr}$ mice were transcardially perfused with 0.1 M PB containing 4% PFA and 0.1% glutaraldehyde, and brains were post-fixed in 0.1 M PB containing 4% PFA. Floating sections were cut on a vibratome at 50 μm, and stained with rat anti-CD11b and rabbit anti-S100β antibodies with DAPI as described above, except that Triton X-100 was omitted from the blocking buffer. After confocal imaging of a phagocytic astrocyte, the same section was osmium-fixed, dehydrated, and an Epon block was prepared. The area containing the cell was trimmed, and serial ultrathin sections (80 nm) were prepared. The cell was identified and imaged with a scanning electron microscope JSM-7900F (JEOL).

### RNA-seq of isolated astrocytes

Astrocytes were isolated using a MACS system according to our previous study (Komine et al, 2018). The workflow is shown in Appendix Fig S4. Briefly, hippocampi were dissected from two transcardially perfused Siglech$^{dtr/dtr}$ mice 2 d after PBS or DT administration, and then dissociated at 37°C for 15 min using the Neural Tissue Dissociation Kit Postnatal Neurons (Miltenyi Biotec). After removal of myelin debris using Myelin Removal Beads II (Miltenyi Biotec), Fc receptors were blocked with anti-CD16/CD32 antibodies. Our preliminary experiment found that the purity of astrocytes increased after removal of microglia from the cell suspension. Therefore, the blocked cells were first reacted with anti-CD11b MicroBeads (Miltenyi Biotec) and loaded onto an LS column

(Miltenyi Biotec) to deplete microglia. The flow-through was then incubated with anti-astrocyte cell surface antigen-2 (ACSA-2) MicroBeads (Miltenyi Biotec) for astrocyte-specific labeling (Batiuk *et al*, 2017) and loaded onto an LS column. Eluted astrocytes were subjected to total RNA isolation using RNeasy Micro Kit (Qiagen). Three pools of RNA, each of which was prepared from the hippocampi of two mice, were obtained from three independent experiments. Total RNA was quantified using a 2100 Bioanalyzer with an RNA 6000 Pico Kit (Agilent Technologies). Libraries were prepared using the SMARTer Stranded RNA-seq Kit according to the manufacturer's protocol (Clontech Laboratories) and then sequenced with 151-nt paired-end reads on the NovaSeq 6000 system (Illumina).

## Quantification of RNA-seq data

Adapter sequences and low-quality (quality value < 20) bases were trimmed from 3′-ends of the RNA-seq reads with Trim Galore v0.5.0 (https://www.bioinformatics.babraham.ac.uk/projects/trim_galore/). The reads were qualified with FastQC v0.11.8 (https://www.bioinformatics.babraham.ac.uk/projects/fastqc/) before and after trimming. The processed reads were mapped on the mouse genome version mm10 using HISAT2 v2.1.0 (Kim *et al*, 2019) by employing strand specificity information, followed by removal of reads mapped to rDNA regions using BEDTools v2.25.0 (Quinlan & Hall, 2010). The statistics of RNA-seq reads is shown in Appendix Table S1. From the mapping data, expressions of individual genes were quantified with StringTie v1.3.5 (Pertea *et al*, 2015) using the mm10 gene annotation. Analysis of differential expression between the PBS and DT groups was performed with edgeR v3.24.3 (Robinson *et al*, 2010) implemented in R v3.5.1, and the *q*-value was calculated as the false discovery rate for multiple testing correction of the *p*-values by the qvalue package v2.24.1 (Storey & Tibshirani, 2003) in R. Counts per million (CPM) and transcripts per million (TPM) values were computed with StringTie and edgeR, respectively.

## GO analyses

GO biological process analysis and gene set enrichment analysis were performed on significantly upregulated 160 genes (*q* < 0.05), using g:Profiler (https://biit.cs.ut.ee/gprofiler/gost; Raudvere *et al*, 2019) with the KEGG database (Kanehisa *et al*, 2019).

## Primary culture of microglia

Primary microglia were cultured from neonatal brains (Tokizane *et al*, 2017). Cerebra dissected from P0–2 WT mice were minced with a fine blade and disrupted by treatment with 0.25% trypsin and 500 µg/ml DNase I (Sigma-Aldrich). After removal of undispersed tissue using a 70-µm pore cell strainer (Corning), cells were suspended in 10 ml DMEM containing 10% fetal bovine serum (FBS) (Thermo Fisher Scientific) and seeded in poly-*L*-lysine-coated 75-cm$^2$ culture flasks ($0.5 \times 10^7$ cells/flask). The culture was maintained by changing the medium every 2–3 d. After 10–14 d in culture, detached microglia were collected by centrifugation. The purity was > 98%. For fluorescent labeling of membranes, WT microglia were incubated with PKH dyes (PKH67: green color; PKH26: red color) (Sigma-Aldrich) according to the manufacturer's protocol. PKH dye-labeled microglia were exposed to UV for 30 s in

UV illuminator FAS-III (TOYOBO). Our preliminary experiment confirmed that almost all microglia died 10 h after the UV exposure.

## Primary culture of astrocytes

Primary astrocytes were obtained as described previously with slight modification (Schildge *et al*, 2013). Cerebra were dissected from P0–2 WT or CAG-EGFP mice, cut into small pieces using a fine surgical blade, and disrupted by treatment with 0.25% trypsin and 100 µg/ml DNase I (Sigma-Aldrich). After removal of undispersed tissue using a 70-µm pore cell strainer (Corning), cells were suspended in 10 ml DMEM containing 10% FBS (Thermo Fisher Scientific) and seeded in poly-*L*-lysine-coated 75-cm$^2$ culture flasks ($1.0 \times 10^7$ cells/flask). The culture was maintained by changing the medium every 2–3 d. After 10–12 d in culture, microglia were detached from the bottom of the flask by vigorous shaking and discarded. Adhered cells, which mainly consisted of astrocytes, were dispersed by treatment with 0.05% trypsin, seeded on a poly-*L*-lysine-coated culture dish, and maintained in DMEM containing 10% FBS until use. The purity of astrocytes was > 95%.

## Evaluation of siRNA knockdown

Maintained WT astrocytes were dispersed by treatment with 0.025% trypsin, seeded in 12-well culture dishes coated with laminin (Sigma), pre-cultured to ~50% confluence, and transfected with 10 pmol siRNA (Silencer Select siRNA; Thermo Fisher Scientific) using 3 µl Lipofectamine RNAiMAX (Thermo Fisher Scientific) according to the manufacturer's protocol. siRNAs were as follows: negative control (#4390847), siTyro3 (#s75638 and #s75639), siAxl (#s76975 and #s76976), siMertk (#s69802 and #s69803), and siMegf10 (#s88802 and #s88804). Astrocytes were then processed for Western blotting (for siAxl, siMertk, and siMegf10) or qPCR (for siTyro3) 48 h after transfection.

## Co-culture of astrocytes with apoptotic microglia

Maintained astrocytes were dispersed by treatment with 0.025% trypsin. For immunocytochemistry, CAG-EGFP mouse-derived astrocytes were seeded on laminin-coated glass coverslips placed on the bottom of 24-well culture dishes, and pre-cultured to ~50% confluence in DMEM containing 10% FBS. PKH26-labeled apoptotic microglia were added to the dishes ($2.5 \times 10^4$ cells/well) and co-cultured with astrocytes for 10–12 h. The astrocytes were then processed for immunocytochemistry after vigorous shaking of the dishes to remove microglial debris attached to cells. For flow cytometry, WT mouse-derived astrocytes were seeded on laminin-coated 12-well culture dishes, pre-cultured to ~50% confluence, and transfected with 10 pmol siRNA as described above. After 48 h, PKH67-labeled apoptotic microglia were added to the dishes ($2.5 \times 10^4$ cells/well) and co-cultured with astrocytes for 10–12 h. The astrocytes were then processed for flow cytometry after vigorous shaking of the dishes to remove microglial debris attached to cells.

## Immunocytochemistry

Co-cultured astrocytes and PKH26-labeled apoptotic microglia on glass coverslips were fixed with 4% PFA. The samples were washed

in PBS, treated with blocking solution (PBS containing 1% BSA, 0.1% Triton X-100, and 0.1% NaN₃) and reacted with goat anti-GFAP or rat anti-LAMP-1 antibodies (Santa Cruz Biotechnology #sc-19992, RRID: AB_2134495) diluted in the blocking solution at 4°C overnight. After washing in PBS, the samples were reacted with a secondary antibody conjugated with Alexa Fluor 647 at room temperature for 1 h and then washed in PBS. The coverslips were mounted on glass slides, and then, images were acquired using an FV10i confocal microscope.

### Western blotting

Hippocampi of adult WT mice were homogenized in lysis buffer containing 40 mM Tris base, 8 M urea, and 2% CHAPS (Konishi *et al*, 2006). After centrifugation at $10,000 \times g$ for 20 min at 4°C, the supernatant was obtained and diluted with an equal volume of 2× Laemmli sample buffer. Cultured astrocytes were lysed directly with 1× Laemmli sample buffer 48 h after siRNA transfection. Proteins were separated by SDS–PAGE and transferred to polyvinylidene difluoride membranes (Merck Millipore). After blocking with blocking solution (Tris-buffered saline/0.1% Tween-20 [TBST] containing 5% skimmed milk or 2.5% BSA), membranes were incubated with rabbit anti-Tyro3 (Cell Signaling Technology #5585, RRID: AB_10706782), goat anti-Axl (R&D Systems #AF854, RRID: AB_355663), goat anti-Mertk (R&D Systems #AF591, RRID: AB_2098565), rabbit anti-MEGF10 (Merck Millipore #ABC10, RRID: AB_11204003), or anti-glyceraldehyde-3-phosphate dehydrogenase (GAPDH) (Thermo Fisher Scientific #AM4300, RRID: AB_2536381) antibodies diluted in blocking solution. Membranes were then washed in TBST and then incubated with secondary antibodies conjugated with horseradish peroxidase (GE Healthcare) in TBST. Signals were visualized using ECL Prime (GE Healthcare) and a luminescent image analyzer, LAS-4010 (GE Healthcare). Band intensity was determined using Image J software (version 10.2, NIH).

### Statistical analyses

RNA-seq data were analyzed with edgeR (Bioconductor), and other data were assessed using Prism 8 (GraphPad Software) or SPSS (version 24, IBM). Values are expressed as the mean ± SEM. Two-tailed unpaired Student's *t*-test was used for comparison of two groups. For non-parametric data, the Mann–Whitney *U*-test was used. To compare > 3 groups, one-way analysis of variance (ANOVA) with repeated-measures, one-way ANOVA with post hoc Tukey's test, or the Kruskal–Wallis test with *post hoc* Dunn's test (for non-parametric data) was used. $P$ or $q < 0.05$ was considered statistically significant.

## Data availability

The RNA-seq data set has been deposited in NCBI GEO database (No.: GSE142022; http://www.ncbi.nlm.nih.gov/geo/query/acc.cgi?acc=GSE142022). The data supporting the findings are available from the corresponding author (HKo) upon request.

**Expanded View** for this article is available online.

### Acknowledgements

This work was supported by KAKENHI [Grants-in-Aid for Scientific Research on Priority Areas "Scrap & Build" 17H05743, "Grant-in-Aid for Scientific Research (B)" 16H05117, "Grant-in-Aid for Scientific Research (B)" 19H03395 to HKi, "Grant-in-Aid for Scientific Research (C)" 16K07055, and "Grant-in-Aid for Scientific Research (C)" 19K06904 to HKo] from the Ministry of Education, Culture, Sports, Science and Technology (MEXT) of Japan and by grants from the Nasu Foundation to HKo. We are grateful to Dr. P. Crocker (University of Dundee) for the anti-Siglec-H antibody, Dr. H. Ohnishi (Gunma University) for mouse transfer, Ms. Y. Itai and N. Tawarayama and K. Muraki for technical assistance, and Ms. A. Asano for secretarial work. We also acknowledge the Division for Medical Research Engineering, Nagoya University Graduate School of Medicine, for use of a flow cytometer, biomolecular imager, and 3D reconstruction software. We thank Dr. J. Allen from Edanz Group (www.edanzediting.com/ac) for editing a draft of this manuscript.

### Author contributions

HKo and HKi designed the study. HKo, TOk, YH, OK, HT, MM, FO, MK, AN, and MT performed the experiments. HKo, YH, OK, HT, MM, FO, YK, KY, TOg, and HKi analyzed the data. TTak, NU, SJ, KO, TTam, and KS provided the materials. HKo, YH, and HKi wrote the manuscript.

### Conflict of interest

The authors declare that they have no conflict of interest.

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
