## [Review Process File · The EMBO Journal]

Astrocytic phagocytosis is a compensatory mechanism for microglial dysfunction

Hiroyuki Konishi, Takayuki Okamoto, Yuichiro Hara, Okiru Komine, Hiromi Tamada, Mitsuyo Maeda, Fumika Osako, Masaaki Kobayashi, Akira Nishiyama, Yosky Kataoka, Toshiyuki Takai, Nobuyuki Udagawa, Steffen Jung, Keiko Ozato, Tomohiko Tamura, Makoto Tsuda, Koji Yamanaka, Tomoo Ogi, Katsuaki Sato, and Hiroshi Kiyama

DOI: [10.15252/embj.2020104464](https://doi.org/10.15252/embj.2020104464)

Corresponding authors: [Hiroyuki Konishi \(konishi@med.nagoya-u.ac.jp\)](mailto:konishi@med.nagoya-u.ac.jp) , [Hiroshi Kiyama \(kiyama@med.nagoya-u.ac.jp\)](mailto:kiyama@med.nagoya-u.ac.jp)

Review Timeline:

Submission Date:	13th Jan 20
Editorial Decision:	13th Feb 20
Revision Received:	28th Jul 20
Editorial Decision:	13th Aug 20
Revision Received:	16th Aug 20
Accepted:	18th Aug 20

Editor: *Karin Dumstrei*

Transaction Report:

Dear Dr. Konishi,

Thank you for submitting your manuscript to The EMBO Journal. Your study has now been seen by two referees and their comments are provided below.

As you can see the referees appreciate the added insight and are supportive of publication here. They raise a number of relevant concerns that are clearly outlined below. Should you be able to address the concerns raised then I would like to invite you to submit a revised manuscript. I should add that it is EMBO Journal policy to allow only a single round of major revision and that it is therefore important to address the raised concerns at this stage.

Let me know if we need to discuss any points in further details. Happy to do so!

When preparing your letter of response to the referees' comments, please bear in mind that this will form part of the Review Process File, and will therefore be available online to the community. For more details on our Transparent Editorial Process, please visit our website:

<https://www.embopress.org/page/journal/14602075/authorguide#transparentprocess>

Thank you for the opportunity to consider your work for publication. I look forward to your revision.

with best wishes

Karin

Karin Dumstrei, PhD
Senior Editor
The EMBO Journal

Further information is available in our Guide For Authors:

The revision must be submitted online within 90 days; please click on the link below to submit the revision online before 13th May 2020.

Link Not Available

Referee #1:

The authors describe the compensatory capacity of non-professional phagocytes of CNS cells (ie astrocytes), to remove cellular debris under conditions where the microglia phagocytosis is defective. They employ a microglia-specific depletion model by administering diphtheria toxin (DT) in *Siglechdtr/dtr* mice. They use histological approaches to evaluate the resulting microglia debris accumulation and clearance in response to astrocytic activation. They conclude that the clearance is not mediated by infiltrated monocytes, surviving microglia or perivascular macrophages, but rather by astrocytes. RNAseq analysis of isolated astrocytes revealed a unique gene signature that combines features from previously described neurotoxic and neuroprotective astrocytic activation. Furthermore, they address in vitro the capacity of astrocytes to phagocyte apoptotic microglia through the TAM receptors. Finally, they analyze astrocytic phagocytosis in genetically modified mice (*Cx3cr1CreER/CreER*, *Csf1op/op*, *Tyrobp^{-/-}*, *Irf8^{-/-}*) that display microglia impairments. This is an excellent study that is well performed, well described and very important for the field. However, several points still needs to be addressed.

Comments

1. The authors focused only on the hippocampal regions. In order to draw general conclusions of

astrocytes biology in the absence of microglia other CNS regions should be analyzed as well.

2. There are no A1 or A2 astrocytes existing in nature. This oversimplification should be avoided and the respective reference should be omitted.

3. The authors should provide nuclear counterstaining in their images, especially in those that they use to quantify cell numbers and those that show depletion of microglia cells. The overall nuclear staining provides a more complete picture of the parenchymal changes.

4. Please clarify what are considered as microglia debris. Are there apoptotic nuclei?

5. In many cases, the authors provided only qualitative assessment of the several features of their model. Quantitative analysis of such features, assist the reader on evaluating the robustness of the phenotype and potential variance between the individual samples in this animal model.

- Figure 1b, the authors comment on the morphological differences of microglia, without providing any further analysis apart from representative images.

- Supplementary Figure 2d, they comment on proliferation of the surviving microglia as a way to replenish their numbers. Are all remaining microglia proliferative?

- Figure 2b, please include the quantification graph in the figure.

- Figure 2c, please provide a quantification of the number of perivascular macrophages in the PBS and DT group.

6. In Figures 1 and 2 the authors define the microglia debris by the Iba1 puncta in the tissue. Then in Figure 4 they start using CD11b to target them, stating that "..., we used an antibody against CD11b, instead of Iba1, because our preliminary experiment found that microglial debris was detected more clearly by staining for CD11b than Iba1." The authors should provide a co-immunostaining for Iba1, CD11b together with a pan-nuclear staining, comparing the PBS, 2days and 4days after DT administration.

7. For Figure 2:

- For the Siglechdtr/dtr:Ccr2RFP/+ analysis please provide in addition data from the 4days post DT administration, when the microglia debris are removed (as shown in Figure 1) to estimate whether delayed recruitment is taking place.

- Is the blood-brain barrier permeability affected in the Siglechdtr/dtr mice after administration of DT?

8. For Figure 4:

- 4b,c,d: The authors represent the data as number of debris per one astrocyte. How did the authors define the single cell by the S100 β immunostaining? In the representative image the S100 β signal is not uniformly distributed along the cell. There is high number of punctate signal around the main cell body. How do the authors differentiate the branches of the cell of interest from neighboring cells? In Figure 4a the interactions are clearer.

- 4c: How do the authors define the "contacted debris". Please indicate on the representative image.

- 4e: How do the authors recognize that the delineated area is part of an astrocyte and how do they know that the internalized structure is microglia debris? Did they perform prior to imaging any immunolabelling for astrocytes and microglia prior to electron microscopy?

- Apart from the interaction and internalization of the microglia debris by astrocytes, is there any upregulation/activation of the lysosomal machinery in astrocytes in vivo?

9. For Figure 5:

- Please provide data for the purity of the sorted astrocytic population

- Please specify the total number of mice used per group for the RNAseq analysis. In the Materials and Methods under "RNA-seq of isolated astrocytes" it is stated that hippocampi from 2 mice were used. Is this correct?

- Do the expression levels of Tyro3/Axl/Mertk change in protein level on astrocytes in the Siglechdtr/dtr DT treated mice?

10. For Figure 6:

- 6c,f,g: In the flow analysis of internalized apoptotic microglia, how do the authors know that the apoptotic cells are indeed inside the GLAST+ astrocytes? If the apoptotic microglia are attached to the surface and not internalized, the GLAST+ population will still be positive for the PKH dye used to label the microglia cells. As a result, any changes in "internalized microglia debris" that are detected in the different knockdown of the receptors could be attributed to the reduced levels of the respective proteins on the cell surface that in turn bind on fewer apoptotic cells and not changes in phagocytosis. Indeed, in Figure 6b (left panel) it seems that there are microglia debris that do not co-localize with the LAMP1 signal.

11. For Figure 7: Please comment why 3week old mice were used instead of adult 8-16weeks old (as in the rest of the experiments)

- 7b,c: The images are in low resolution. The authors should use the same marker across all the groups for microglia quantification, if they are to compare their numbers across the groups. Please provide representative images for the Cxc3cr1CreER/CreER and Tyrobp-/- . In addition, all previous analyses were performed on the CA1 region of the hippocampal formation, yet here they provide data from the cerebral cortex. The authors should be consistent on the brain regions that they evaluate.
- 7d,e,f: The authors evaluate the interactions between microglia and astrocytes with spontaneous apoptotic cells in the brain. As they state these events are quite rare in the post-developmental brain. Please provide the absolute numbers of the apoptotic cells detected per mouse brain across the groups. Did the authors investigate the response of astrocytes in regions where apoptosis is higher such as the neurogenic niches (subgranular zone of dentate gyrus, subventricular zone). Are the astrocytic phagocytic features present across the different groups?

12. Supplementary Figure 4: The authors show that the heterozygous and homozygous Siglechdtr mice show reduction of the Siglec-H expression both in mRNA and protein level in the non-injected mice. Does this lack of Siglec-H affect the microglia numbers and functionality, which could potential lead to an activated astrocytic phenotype even in the absence of the DT-mediated microglia depletion? In addition, the authors state that "As with homozygotes, astrocytes interacted with microglial debris in the hippocampal CA1 of heterozygotes (Supplementary Fig. 4c), indicating that this contact did not result from Siglech deficiency.". They only provide one representative image of one GFAP+ cell interacting with a CD11b+ structure. Please provide the same analysis as in Figure 4b,c,d.

Referee #2:

Konishi et al. showed that in the absence of microglia or the presence of dysfunctional microglia, astrocytes will adapt a phagocytic role to engulf microglial debris in the CA1 region of hippocampus. Using Siglech DTR mice and parabiosis model, they demonstrated that astrocytes and not infiltrating monocytes are responsible for debris clearance as no GFP+ monocytes were found in the hippocampal parenchyma. RNAseq of hippocampal astrocytes revealed increased expression of both A1 and A2 reactive astrocyte markers correlating with increased GFAP expression and hypotrophy seen by IHC. In addition, RNAseq of these cells showed they expression of TAM

phagocytic receptors in both healthy and diseased states. Specifically, Mertk and Axl are required in vitro for the engulfment of microglial debris. Accumulation of cellular debris can be detrimental to function and recovery, this paper provides evidence for a compensatory phagocytic mechanism in the absence of microglia. Parallel roles of astrocytes and microglia in engulfment of synapses have been extensively characterized. Therefore, the discovery of astrocyte compensation for microglia engulfment is expected. The concept itself is not novel but it is nice to demonstrate it with rigorous experiments. Addressing the following questions may improve the manuscript:

1. Do astrocytes compensate for microglia engulfment of synapses during development?
2. The purity of astrocytes from MACS isolation need to be described. This is especially important since much of reactive astrocyte genes are also expressed by activated microglia. How to know the induced genes are from astrocytes instead of microglia contamination?
3. How astrocyte engulfed microglia components were identified as such in EM was not described. How frequently these events were observed by EM in microglia depletion vs wt mice need to be described.
4. RNAseq sequencing depth needs to be described.
5. In figure 4A PBS treated you can see how many cd11b processes are surrounding/potentially interacting with astrocytes. How can you be certain that astrocytes are extending their process out to a microglia debris? How can you be sure it wasn't closely associated when alive and it happen to die next to the astrocyte and the astrocyte process are just surrounding it?
6. it seems like other isoforms of Axl and Mertk are still present in Figure 5D; are they important/functional and could they explain the minimal reduction in engulfment capabilities
7. Irf8 KO mice: Surrounding debris does not equal engulfment; need to show the debris is in the lysosome or phagocytic cup to make this claim.
8. When describing their RNA-seq data, the authors fail to mention any pathways or genes that are downregulated; an explanation as to why no downregulated genes were examined would be helpful.

Referee #1:

The authors describe the compensatory capacity of non-professional phagocytes of CNS cells (ie astrocytes), to remove cellular debris under conditions where the microglia phagocytosis is defective. They employ a microglia-specific depletion model by administering diphtheria toxin (DT) in *Siglechdtr/dtr* mice. They use histological approaches to evaluate the resulting microglia debris accumulation and clearance in response to astrocytic activation. They conclude that the clearance is not mediated by infiltrated monocytes, surviving microglia or perivascular macrophages, but rather by astrocytes. RNAseq analysis of isolated astrocytes revealed a unique gene signature that combines features from previously described neurotoxic and neuroprotective astrocytic activation. Furthermore, they address in vitro the capacity of astrocytes to phagocytose apoptotic microglia through the TAM receptors. Finally, they analyze astrocytic phagocytosis in genetically modified mice (*Cx3cr1CreER/CreER*, *Csf1op/op*, *Tyrobp^{-/-}*, *Irf8^{-/-}*) that display microglia impairments.

This is an excellent study that is well performed, well described and very important for the field. However, several points still needs to be addressed.

Comment #1

The authors focused only on the hippocampal regions. In order to draw general conclusions of astrocytes biology in the absence of microglia other CNS regions should be analyzed as well.

According to the comment, we examined other brain regions. In all areas examined, we consistently observed phagocytosis of microglial debris by astrocytes concomitant with GFAP upregulation. We have shown data of representative brain regions (cerebral cortex, thalamus and medulla) in Figure EV3, and have described the results in the Results section (Line 211–213).

Comment #2

There are no A1 or A2 astrocytes existing in nature. This oversimplification should be avoided and the respective reference should be omitted.

We agree with this comment because our experimental model is not a pathological one. We have deleted the previous Figure 5b, which was a heatmap image showing gene

expression levels of A1 and A2 marker molecules, and the corresponding sentences in the Results and Discussion sections. We have highlighted throughout the manuscript that the activated astrocytes exhibited a pro-inflammatory gene expression profile (Abstract section: Line 52) (Results section: Line 248 and 259-268) (Discussion section: Line 415-423).

Comment #3

The authors should provide nuclear counterstaining in their images, especially in those that they use to quantify cell numbers and those that show depletion of microglia cells. The overall nuclear staining provides a more complete picture of the parenchymal changes.

According to this comment together with Comments #4 and #6 of this reviewer, we re-examined several immunohistochemical studies by adding DAPI staining.

1. *Figure 1*: (The number of microglia and microglial debris pieces in our microglia ablation model).

In this revision, we divided Iba1⁺ microglia into three categories according to nuclear morphology (live microglia with a normal nucleus; apoptotic microglia with a pyknotic or fragmented nucleus; pieces of microglial debris [spheres with a diameter > 2 μm] with no nucleus) and re-quantified changes in three microglial categories over time. We have renewed Figure 1, and have indicated the categorization in its legend (Line 1082–1091). We have also rewritten the corresponding results in the Results section (Line 130–137 and 155–160) and the quantification method in the Materials and Methods section (Line 536–545). This comment improved our data without subjective bias.

2. *Figure EV2D and EV2E*: (Ki-67⁺ proliferating microglia after microglial ablation).

We re-performed Ki-67/Iba1 immunohistochemistry with the addition of DAPI staining (Figure EV2D), and have added quantified data (Figure EV2E). We have also added a description in the Results section (Line 150 and 151).

3. *Figure EV3A, D and G*: (Microglial ablation and astrocyte activation in our microglia ablation model in brain regions other than the hippocampus).

All these immunostainings are now also accompanied by DAPI staining.

4. *Figure EV4A and D*: (Microglial ablation and astrocyte activation in other models of microglial ablation).

DAPI staining was added.

5. *Appendix Figure S2* (Immunostaining of microglial debris with anti-CD11b or Iba1 antibodies).

DAPI staining was added.

Comment 4

Please clarify what are considered as microglia debris. Are there apoptotic nuclei?

As described in the response to Comment #3 of this reviewer, we define Iba1⁺ spheres (> 2 μm in diameter) with no nucleus as pieces of microglial debris. We have described this definition in the Results section (Line 155 and 156), Materials and Methods section (Line 542 and 543) and the legend of Figure 1 (Line 1090 and 1091).

Comment #5

In many cases, the authors provided only qualitative assessment of the several features of their model. Quantitative analysis of such features, assist the reader on evaluating the robustness of the phenotype and potential variance between the individual samples in this animal model.

We appreciate this comment and, accordingly, we have added quantitative analysis where necessary (Figure 1B, 2B, 2D, and 7F) (Figure EV2E) (Appendix Figure S1, S3E, S3F, and S7). We think the reliability of our data has been improved.

Comment #5-1

Figure 1b, the authors comment on the morphological differences of microglia, without providing any further analysis apart from representative images.

According to this comment, we performed quantitative morphometric analyses of microglia at 0, 2, 4, 7 and 28 days after DT treatment. The results are shown in Appendix Figure S1: microglia exhibited less ramified morphology in the 2–7 days post DT administration; however, the morphology became identical to that before DT administration by day 28. We have described these results in the Results section (Line 134–137, 157, and 158) and the Discussion section (Line 382 and 383), and have added the method to the Materials and Methods section (Line 546–554).

Comment #5-2

Supplementary Figure 2d, they comment on proliferation of the surviving microglia as a way to replenish their numbers. Are all remaining microglia proliferative?

Supplementary Figure 2 in the previous version is now Figure EV2 in the revised version. According to the comment, we performed quantitative analysis and have added a graph showing the percentage of Ki-67⁺ microglia at 0, 2, 4, 7 and 28 days after DT treatment (Figure EV2E). On day 4, the onset of the recovery phase, ~90% of microglia were proliferative. We have described this in the Results section (Line 150 and 151), and have added the quantification method to the Materials and Methods section (Line 536–545).

Comment #5-3

Figure 2b, please include the quantification graph in the figure.

In response to this comment, we have included a quantification graph in Figure 2B.

Comment #5-4

Figure 2c, please provide a quantification of the number of perivascular macrophages in the PBS and DT group.

We quantified the number of perivascular macrophages in the PBS and DT groups, and have shown the result in the newly added Figure 2D. The cell number was unchanged by DT treatment, indicating that perivascular macrophages are insensitive to DT and appear intact, in contrast to microglia. We have shown the result in the Results section (Line 187 and 188), and have added the quantification method to the Materials and Methods section (Line 555–563).

Comment #6

In Figures 1 and 2 the authors define the microglia debris by the Iba1 puncta in the tissue. Then in Figure 4 they start using CD11b to target them, stating that "... we used an antibody against CD11b, instead of Iba1, because our preliminary experiment found that microglial debris was detected more clearly by staining for CD11b than Iba1." The authors should provide a co-immunostaining for Iba1, CD11b together with a pan-nuclear staining, comparing the PBS, 2days and 4days after DT administration.

This comment is addressed in Line 207 and 208 in the revised version of the manuscript. We have added a new figure (Appendix Figure S2) to demonstrate that microglial debris was clearly detected by CD11b immunostaining.

Comment #7

For Figure 2:

- **For the *Siglechdtr/dtr*:*Ccr2RFP/+* analysis please provide in addition data from the 4days post DT administration, when the microglia debris are removed (as shown in Figure 1) to estimate whether delayed recruitment is taking place.**

This comment is appropriate and helpful to improve the present study. We tested this possibility, and demonstrated that no RFP⁺ monocytes infiltrated the hippocampal parenchyma at day 4 as well as day 2. We have added the day 4 result to Figure 2A (the bottom column). We also modified sentences in the Results (Line 176) and Discussion sections (Line 397).

- **Is the blood-brain barrier permeability affected in the *Siglechdtr/dtr* mice after administration of DT?**

In our preliminary experiment, blood-brain barrier breakdown, which was determined by Evans blue extravasation, did not occur in *Siglech*^{dtr/dtr} mice after DT treatment, although astrocytes were activated and their gene expression profile was changed. We think that this result is not relevant to the main focus of this manuscript; therefore we have not shown this data.

Comment #8

For Figure 4:

- **4b,c,d: The authors represent the data as number of debris per one astrocyte. How did the authors define the single cell by the S100 β immunostaining? In the representative image the S100 β signal is not uniformly distributed along the cell. There is high number of punctate signal around the main cell body. How do the authors differentiate the branches of the cell of interest from neighboring cells? In Figure 4a the interactions are clearer.**

The previous Figure 4b, c and d are now presented as Figure 4C, D and E, respectively. We had not written the quantification method in detail. We used S100 β instead of

GFAP for the quantification because the entire astrocyte morphology was visualized more clearly in using S100 β , as described in Line 223–225. Confocal microscopy images of a randomly selected single S100 β ⁺ astrocyte, whose cell body was entirely present within a 30 μ m-thick section, were taken and analyzed. Because the cell morphology was 3D reconstructed by cell surface rendering using *Imaris* software, processes of the single astrocyte were distinguished from those of neighboring astrocytes. We have described the method in detail in the Materials and Methods section (Line 567–574), and have improved the legend of Figure 4D and E (Line 1153–1158).

• 4c: How do the authors define the "contacted debris". Please indicate on the representative image.

The previous Figure 4c is now presented as Figure 4D. Another reviewer also commented on the method. We had not described the precise definition in detail in the previous version of the manuscript. We defined CD11b⁺ spheres (diameter > 0.5 μ m), whose surface was more than 30% covered by astrocyte processes, as contacted debris in 3D reconstructed images. We show a representative image of phagocytic astrocytes in Figure 4C, whose insets demonstrate examples of debris with approximately 30% or 50% coverage. Accordingly, we have changed the graph title of Figure 4D to “Number of contacted debris (coverage > 30%)/astrocyte”. We have also modified the figure legend (Line 1146–1152), and the method in the Materials and Methods section (Line 567–574). Appendix Figure S3E is newly added to show the case of *Siglech*^{dtr/+} heterozygotes, according to Comment #12 of this reviewer, and the quantification method was the same.

• 4e: How do the authors recognize that the delineated area is part of an astrocyte and how do they know that the internalized structure is microglia debris? Did they perform prior to imaging any immunolabelling for astrocytes and microglia prior to electron microscopy?

Electron microscopy (EM) analysis (Figure 4e in the previous version) is included in Figure 4B in this revised version. Another reviewer also mentioned this point. In EM images, astrocytes generally have relatively clearer cytoplasm because of low electron density. We have extensive experience in the observation of brain tissue using EM, and we can differentiate cell types by their morphology and brightness of EM images. In our

previous EM study, we found that astrocyte cytoplasm, which was relatively bright, occasionally contained degenerated materials with high electron density only after microglial ablation. Together with the light microscopy observations, we presumed the degenerated material to be microglial debris; however, there was no direct evidence. To obtain such direct evidence, we performed correlative light and electron microscopy (CLEM). After immunohistochemical identification of a phagocytic astrocyte, which engulfed CD11b⁺ microglial debris, the same astrocyte was subsequently analyzed by electron microscopy. The result demonstrated that astrocytes engulfed degenerated microglial components. We have added the result of CLEM analysis to Figure 4B using three images from light and electron microscopy. We have also presented this data in the Results section (Line 225–230) and have updated the EM method in the Materials and Methods section (Line 677–686).

• Apart from the interaction and internalization of the microglia debris by astrocytes, is there any upregulation/activation of the lysosomal machinery in astrocytes in vivo?

Gene ontology (GO) analysis of upregulated genes in RNA-seq data demonstrated no alteration of lysosome-related biological terms, although the GO resource contains > 100 lysosome-related terms, such as “lysosome organization”. This suggests that astrocytes are pre-equipped with phagocytosis machinery for the clearance of microglial debris. We have added this description to the Results section (Line 273–277).

Comment 9

For Figure 5:

• Please provide data for the purity of the sorted astrocytic population

Another reviewer also mentioned this point. We used a magnetic-activated cell sorting (MACS) system to isolate hippocampal astrocytes, according to our previous study in which spinal astrocytes were isolated (Komine et al, *Cell Death Differ*, 2018). There are two key points in our method. First, for isolation of astrocytes, we used anti-astrocyte cell surface antigen-2 (ACSA-2) magnetic beads (Kantzer et al, *Glia*, 2017), whose utility has been demonstrated in recent papers (Shinozaki et al, *Cell Rep*, 2017; Göbel et al, *Cell Metab*, 2020; Zhang et al, *Proc Natl Acad Sci*, 2020). Second, prior to the reaction with anti-ACSA-2 magnetic beads, microglia were depleted from the cell

suspension using anti-CD11b magnetic beads because our preliminary experiment showed that a small but significant number of microglia contaminated the ACSA-2⁺ fraction when we used anti-ACSA-2 magnetic beads only. In this revision, we checked the purity of astrocytes by flow cytometry using an anti-GFAP antibody, and a representative result is shown in the newly added Appendix Figure S5. The percentage of GFAP⁺ cells in the ACSA-2⁺ fraction was almost 90%, which may be underscored because a minor population of hippocampal astrocytes are GFAP⁻ (Walz and Lang, *Neurosci Lett*, 1998; Jinno, *Neuroscience*, 2011).

Related to this comment, another reviewer also considered the possibility that the activated pro-inflammation pathway in the RNA-seq data (Fig. 5A) results from microglial contamination. However, taking the flow cytometry data (Appendix Figure S4) and newly added RNA-seq data showing the level of microglia marker genes (Appendix Figure S5) into consideration, microglial contamination was very slight, indicating that pro-inflammatory changes did occur in astrocytes. We have described this explanation in the Results section (Line 249, 250, 254–256, and 265–268), and have added the method to the Materials and Method section (Line 636–643, and 690).

• Please specify the total number of mice used per group for the RNAseq analysis. In the Materials and Methods under "RNA-seq of isolated astrocytes" it is stated that hippocampi from 2 mice were used. Is this correct?

We apologize for the poorly written method. We analyzed three pools of RNA, each of which was prepared from the hippocampi of two mice, per group. We have rewritten the sentence in the Materials and Methods section (Line 702 and 703).

• Do the expression levels of Tyro3/Axl/Mertk change in protein level on astrocytes in the Siglechdtr/dtr DT treated mice?

We tried immunohistochemical detection of Axl/Mertk protein using commercial antibodies. However, none of the antibodies tested gave appropriate signals even in microglia, which highly express Axl/Mertk. We assume that the protein level of Axl/Mertk is unchanged in astrocytes after DT treatment, similar to their mRNAs.

Comment #10

For Figure 6:

• 6c,f,g: In the flow analysis of internalized apoptotic microglia, how do the authors know that the apoptotic cells are indeed inside the GLAST+ astrocytes? If the apoptotic microglia are attached to the surface and not internalized, the GLAST+ population will still be positive for the PKH dye used to label the microglia cells. As a result, any changes in "internalized microglia debris" that are detected in the different knockdown of the receptors could be attributed to the reduced levels of the respective proteins on the cell surface that in turn bind on fewer apoptotic cells and not changes in phagocytosis. Indeed, in Figure 6b (left panel) it seems that there are microglia debris that do not co-localize with the LAMP1 signal.

We had considered this possibility in our immunocytochemical and flow cytometric analyses of cultured astrocytes and we performed these experiments with great caution. After co-culture of astrocytes with apoptotic microglia, we shook culture dishes vigorously to remove cell-surface attached microglial debris, resulting in scarcely no microglial debris on the surface of astrocytes on immunocytochemical analysis. We have added this methodological point to the Materials and Methods section (Line 779, 780, and 784–786). Regarding colocalization of microglial debris (red) with lysosomes (cyan) in Figure 6B, the previous low magnification image was misleading, as the reviewer mentioned. We have added a high magnification image as an inset in Figure 6B. The high magnification image clearly shows that microglial debris almost entirely merged with lysosomes. We have modified the description in the Results section (Line 299–302).

Comment #11

For Figure 7: Please comment why 3week old mice were used instead of adult 8-16weeks old (as in the rest of the experiments)

The reason was briefly written in the previous version of the manuscript, but may not have been sufficient explanation. We have described the reason more precisely in the Results section (Line 339–344). The reason is as follows. In older mice, the identification of apoptotic cells on tissue sections is very hard; however, in neonatal and younger mice, spontaneous apoptotic cells are easily detectable (Oppenheim, *Annu Rev Physiol*, 2017). In fact, neonatal brain contains many spontaneous apoptotic cells; however, glial cells appear to be very immature. We thus considered that the neonatal

brain was not suitable for this study. Our preliminary immunohistochemical experiment indicated that the cerebral cortex of 3-week-old mice contained a small but consistent number of cleaved caspase-3⁺ apoptotic cells.

• 7b,c: The images are in low resolution. The authors should use the same marker across all the groups for microglia quantification, if they are to compare their numbers across the groups. Please provide representative images for the Cxc3cr1CreER/CreER and Tyrobp^{-/-}. In addition, all previous analyses were performed on the CA1 region of the hippocampal formation, yet here they provide data from the cerebral cortex. The authors should be consistent on the brain regions that they evaluate.

For the analysis of spontaneous apoptotic cells, we carefully selected the cerebral cortex from several candidate brain regions for several reasons.

1. The region consistently contains natural apoptotic cells.
2. The region contains the same number of microglia in different mouse strains.
3. Cell density is not high. In high cell density regions, such as neurogenic niches, glial interaction with spontaneous apoptotic cells is difficult to analyze, as described in our response to the next comment of this reviewer.

As the reviewer suggests, it is better to analyze the hippocampal CA1 region throughout our study. We therefore tested hippocampal CA1; however, it was very hard to identify spontaneous apoptotic cells.

As for microglial markers to quantitatively study microglial numbers (Figure 7B) (Line 346–350), it is difficult to use the same marker across all mutant strains. This is because Iba1 expression is very low in *Irf8^{-/-}* microglia. We had described this reason in the previous version of the manuscript. To demonstrate this fact, we have added a new figure (Appendix Figure S6). Iba1 expression is almost below the level of detection in *Irf8^{-/-}* microglia, whereas CD11b is clearly detectable in *Irf8^{-/-}* as well as in WT microglia. We have also improved the resolution of immunohistochemical images (Figure 7B).

• 7d,e,f: The authors evaluate the interactions between microglia and astrocytes with spontaneous apoptotic cells in the brain. As they state these events are quite rare in the post-developmental brain. Please provide the absolute numbers of the

apoptotic cells detected per mouse brain across the groups. Did the authors investigate the response of astrocytes in regions where apoptosis is higher such as the neurogenic niches (subgranular zone of dentate gyrus, subventricular zone). Are the astrocytic phagocytic features present across the different groups?

In response to this comment, we have shown the absolute number of spontaneous apoptotic cells per section in the newly added Appendix Figure S7. There are no statistical differences among mouse strains. We have added this description to the Results section (Line 356–358).

As the reviewer suggests, neurogenic niches are candidate regions for the analysis. However, preliminary experiments indicated that neurogenic regions were unsuitable for our study. We occasionally found spontaneous apoptotic cells in neurogenic regions. However, cell density was very high in those regions, and the paths of glial processes were limited to narrow intercellular spaces, resulting in frequent “false-positive” interaction of apoptotic cells with astrocyte processes. We have shown an example image below. In this image of the dentate gyrus subgranular zone in WT mice, microglia (red) clearly encircle a spontaneous apoptotic cell (blue), indicating that microglia phagocytose the apoptotic cell. However, astrocyte processes (green) also appear to interact with the apoptotic cell.

Regarding phagocytic features of astrocytes in mutant mice, only WT and *Irf8*^{-/-} mice were quantitatively analyzed in the previous version of the manuscript. We had obtained microscopy data of other mutant strains (*Cx3cr1*^{CreER/CreER}, *Csf1*^{op/op} and *Tyrobp*^{-/-}), but had not analyzed them quantitatively. In this revision, we analyzed them, demonstrating that other mutant strains (*Cx3cr1*^{CreER/CreER}, *Csf1*^{op/op} and *Tyrobp*^{-/-}) are similar to WT mice. We have added the data to Figure 7F. Accordingly, we have modified a sentence in the Results section (Line 363–365).

Comment #12

Supplementary Figure 4: The authors show that the heterozygous and homozygous *Siglech*^{dtr} mice show reduction of the Siglec-H expression both in mRNA and protein level in the non-injected mice. Does this lack of Siglec-H affect the microglia numbers and functionality, which could potential lead to an activated astrocytic phenotype even in the absence of the DT-mediated microglia depletion? In addition, the authors state that "As with homozygotes, astrocytes interacted with microglial debris in the hippocampal CA1 of heterozygotes (Supplementary Fig. 4c), indicating that this contact did not result from *Siglech* deficiency.". They only provide one representative image of one GFAP+ cell interacting with a CD11b+ structure. Please provide the same analysis as in Figure 4b,c,d.

The previous Supplementary Figure 4 is now presented as Appendix Figure S3. Our previous study showed no change of microglial number in the spinal dorsal horn between WT and *Siglech*^{dtr/dtr} mice without DT treatment (Konishi et al, *Glia*, 2017) and we did not examine hippocampal microglia in this study. However, the mRNA level of *Aif1* (the gene encoding Iba1) in hippocampus was the same in PBS-treated WT and *Siglech*^{dtr/dtr} mice (Figure 3A), suggesting that *Siglech* deficiency did not affect microglial number. As for astrocytes, *Gfap* mRNA, whose upregulation is a hallmark of astrocyte activation, was equivalent between PBS-treated WT and *Siglech*^{dtr/dtr} mice (Figure 3A). Therefore, *Siglech* deficiency was unlikely to affect astrocyte activity. Nevertheless, according to the reviewer's suggestion, we performed quantitative analysis of phagocytic activity of astrocytes in *Siglech*^{dtr/+} heterozygotes, and have added the data (Appendix Figure S3E and F). As with *Siglech*^{dtr/dtr} homozygotes, astrocytes contacted and internalized microglial debris in the hippocampal CA1 of heterozygotes, indicating that the engulfment did not result from *Siglech* deficiency. We also have added the description to the Results section (Line 239–246).

Referee #2:

Konishi et al. showed that in the absence of microglia or the presence of dysfunctional microglia, astrocytes will adapt a phagocytic role to engulf microglial debris in the CA1 region of hippocampus. Using Siglech DTR mice and parabiosis model, they demonstrated that astrocytes and not infiltrating monocytes are responsible for debris clearance as no GFP+ monocytes were found in the hippocampal parenchyma. RNAseq of hippocampal astrocytes revealed increased expression of both A1 and A2 reactive astrocyte markers correlating with increased GFAP expression and hypertrophy seen by IHC. In addition, RNAseq of these cells showed they expression of TAM phagocytic receptors in both healthy and diseased states. Specifically, Mertk and Axl are required in vitro for the engulfment of microglial debris. Accumulation of cellular debris can be detrimental to function and recovery, this paper provides evidence for a compensatory phagocytic mechanism in the absence of microglia. Parallel roles of astrocytes and microglia in engulfment of synapses have been extensively characterized. Therefore, the discovery of astrocyte compensation for microglia engulfment is expected. The concept itself is not novel but it is nice to demonstrate it with rigorous experiments. Addressing the following questions may improve the manuscript:

Comment #1

Do astrocytes compensate for microglia engulfment of synapses during development?

Synapse elimination by astrocytic phagocytosis was first demonstrated in 2013 (Chung et al, *Nature*, 2013). Since then, however, very few papers have addressed this astrocyte function *in vivo*. Although the functional difference between microglia and astrocytes in synapse elimination is an interesting issue, the present data do not address this issue. As written in the Discussion section (Line 466–469), we do not discuss this issue further.

Comment #2

The purity of astrocytes from MACS isolation need to be described. This is especially important since much of reactive astrocyte genes are also expressed by

activated microglia. How to know the induced genes are from astrocytes instead of microglia contamination?

Another reviewer also mentioned this point. We used a magnetic-activated cell sorting (MACS) system to isolate hippocampal astrocytes, according to our previous study in which spinal astrocytes were isolated (Komine et al, *Cell Death Differ*, 2018). There are two key points in our method. First, for isolation of astrocytes, we used anti-astrocyte cell surface antigen-2 (ACSA-2) magnetic beads (Kantzer et al, *Glia*, 2017), whose utility has been demonstrated in recent papers (Shinozaki et al, *Cell Rep*, 2017; Göbel et al, *Cell Metab*, 2020; Zhang et al, *Proc Natl Acad Sci*, 2020). Second, prior to the reaction with anti-ACSA-2 magnetic beads, microglia were depleted from the cell suspension using anti-CD11b magnetic beads because our preliminary experiment showed that a small but significant number of microglia contaminated the ACSA-2⁺ fraction when we used anti-ACSA-2 magnetic beads only. In this revision, we checked the purity of astrocytes by flow cytometry using an anti-GFAP antibody, and a representative result is shown in the newly added Appendix Figure S5. The percentage of GFAP⁺ cells in the ACSA-2⁺ fraction was almost 90%, which may be underscored because a minor population of hippocampal astrocytes are GFAP⁻ (Walz and Lang, *Neurosci Lett*, 1998; Jinno, *Neuroscience*, 2011).

As the reviewer mentioned, microglia are thought as the major source of pro-inflammatory molecule, and it is possible that the activated pro-inflammation pathway in the RNA-seq data (Fig. 5A) results from microglial contamination. However, taking the flow cytometry data (Appendix Figure S4) and newly added RNA-seq data showing the level of microglia marker genes (Appendix Figure S5) into consideration, microglial contamination was very slight, indicating that pro-inflammatory changes did occur in astrocytes. We have described this explanation in the Results section (Line 249, 250, 254–256, and 265–268), and have added the method to the Materials and Method section (Line 636–643, and 690).

Comment #3

How astrocyte engulfed microglia components were identified as such in EM was not described. How frequently these events were observed by EM in microglia depletion vs wt mice need to be described.

Another reviewer also mentioned this point. In EM images, astrocytes generally have relatively clearer cytoplasm because of low electron density. We have extensive experience in the observation of brain tissue using EM, and we can differentiate cell types by their morphology and brightness of EM images. In our previous EM study, we found that astrocyte cytoplasm, which was relatively bright, occasionally contained degenerated materials with high electron density only after microglial ablation. Together with the light microscopy observations, we presumed the degenerated material to be microglial debris; however, there was no direct evidence. To obtain such direct evidence, we performed correlative light and electron microscopy (CLEM). After immunohistochemical identification of a phagocytic astrocyte, which engulfed CD11b⁺ microglial debris, the same astrocyte was subsequently analyzed by electron microscopy. The result demonstrated that astrocytes engulfed degenerated microglial components. We have added the result of CLEM analysis to Figure 4B using three images from light and electron microscopy. We have also presented this data in the Results section (Line 225–230) and have updated the EM method in the Materials and Methods section (Line 677–686).

Quantification of the frequency of engulfed microglial debris by EM observation is very difficult. Instead, we performed a quantitative analysis of immunohistochemical images taken by confocal microscopy (Figure 4E).

Comment #4

RNAseq sequencing depth needs to be described.

We show the RNA-seq information, including the number of reads, in the newly added Appendix Table S1 (Line 718). The total reads were ~100,000,000 for each sample. We assume the resolution is high.

Comment #5

In figure 4A PBS treated you can see how many cd11b processes are surrounding/potentially interacting with astrocytes. How can you be certain that astrocytes are extending their process out to a microglia debris? How can you be sure it wasn't closely associated when alive and it happen to die next to the astrocyte and the astrocyte process are just surrounding it?

Another reviewer also commented on the method. We had not written the quantification method in detail in the previous version of the manuscript. We defined CD11b⁺ spheres (diameter > 0.5 μm) as microglial debris in the 3D reconstructed images. Then, the number of microglial debris, whose surface was more than 30% covered by astrocyte processes, was counted. This quantitative analysis was blindly done. There were no microglial debris in control groups (PBS or DT-treated WT mice and PBS-treated *Siglech*^{dtr/dtr} mice) by microscopic observation; however, in the statistical data, there were several contacted debris even in the control groups, indicating that the quantitative analysis was performed without subjective bias. We show a representative image of phagocytic astrocytes in Figure 4C, whose insets demonstrate examples of debris with approximately 30% or 50% coverage. Accordingly, we have changed the graph title of Figure 4D to “Number of contacted debris (coverage > 30%)/astrocyte”. We have also modified the figure legend (Line 1146–1152), and the method in the Materials and Methods section (Line 567–574). Appendix Figure S3E is newly added to show the case of *Siglech*^{dtr/+} heterozygotes, according to a comment of another reviewer, and the quantification method was the same.

Comment #6

It seems like other isoforms of Axl and Mertk are still present in Figure 5D; are they important/functional and could they explain the minimal reduction in engulfment capabilities.

We think this comment should refer to Figure 6D, not 5D. In western blot analysis of Axl and Mertk, we used the same secondary antibody (a peroxidase-conjugated donkey anti-goat IgG antibody). The ~115 kDa band of Axl and Mertk was siRNA-resistant, suggesting that it was a non-specific band resulting from a reaction with the secondary antibody. Indeed, we demonstrated that the ~115 kDa band appeared even when the primary antibodies were omitted. Therefore, in the case of Mertk, the specific band was ~160 kDa, and siRNA knockdown was almost total. For Axl, the ~90 kDa band was specific, and was almost fully downregulated by siRNAs. However, the ~115 kDa Axl band was assumed to a combination of specific and weak non-specific bands. Since the knockdown efficiency of the ~90 kDa band was high, the specific component of the ~115 kDa band was assumed to be efficiently downregulated. The downregulation efficiency of Axl may appear lower, but this is presumably because of the overlap of the

weak non-specific band. We have added a brief description in the legend of Figure 6D (Line 1183–1186). Even in the presence of the overlapping non-specific band, the calculated knockdown efficiency (70–80%) was enough to demonstrate knockdown. We show the summarized image below.

Comment #7

Irf8 KO mice: Surrounding debris does not equal engulfment; need to show the debris is in the lysosome or phagocytic cup to make this claim.

The previous images of *Irf8*^{-/-} mice, particularly the 3D reconstructed image, was ambiguous (Figure 7d in the previous version), as the reviewer mentioned. We have renewed the images. We now show two representative images of *Irf8*^{-/-} mice, one of which clearly demonstrates a spontaneous apoptotic cell enclosed in a phagocytic cup of an astrocyte (the right panels in Figure 7D). We have also added a description to the Results section (Line 362 and 363).

Comment #8

When describing their RNA-seq data, the authors fail to mention any pathways or genes that are downregulated; an explanation as to why no downregulated genes were examined would be helpful.

We had also analyzed downregulated genes, but had not shown the data in the previous version of the manuscript. The reason why we did not show the data was briefly written in the Material and Methods (Page 23, Line 11–15 in the previous version); however, the sentences might not be easily found by readers. To make this point clear, we now show the list of downregulated genes in the newly added Table EV3, and have written the reason in the Results section (Line 254–259). The reason is as follows. Because of

the very slight contamination of microglia in our MACS isolation system, the expression level of microglia-enriched genes was extremely low (Appendix Fig S5). However, their downregulation was very significant because of the decreased microglial number, resulting in the inclusion of many microglia-enriched genes in the list of statistically downregulated genes (Table EV3).

Dear Dr. Konishi,

Thank you for submitting your revised manuscript to The EMBO Journal. Your revision has now been seen by the two referees and their comments are provided below. As you can see both referees appreciate the introduced changes and support publication here.

I am therefore very pleased to let you know that we will accept the manuscript for publication here. Before sending you the formal acceptance letter there are just a few editorial things to sort out.

- Tsuda Makota is missing from the author contribution section
- The appendix file needs to have the individual figures listed in the ToC. Please also add page numbers to the PDF.
- Will you please check that size of the synopsis image is OK. It should be 550 wide by [200-400] high (pixels).
- I have asked our publisher to do their pre-publication checks on the paper. They will send me the file within the next few days. Please wait to upload the revised version until you have received their comments.

Congratulations on a great manuscript!

With best wishes

Karin

Karin Dumstrei, PhD
Senior Editor
The EMBO Journal

- a point-by-point response to the referees' comments, with a detailed description of the changes made (as a word file).
- a word file of the manuscript text.

- individual production quality figure files (one file per figure)
 - a complete author checklist, which you can download from our author guidelines (<https://www.embopress.org/page/journal/14602075/authorguide>).
 - Expanded View files (replacing Supplementary Information)
- Please see out instructions to authors
<https://www.embopress.org/page/journal/14602075/authorguide#expandedview>

The revision must be submitted online within 90 days; please click on the link below to submit the revision online before 11th Nov 2020.

Link Not Available

Referee #1:

The authors have satisfyingly addressed all the comments raised and they have revised their manuscript accordingly. There are no additional comments and the manuscript could be accepted as it is.

Referee #2:

The authors have addressed all my concerns and I therefore support publication of this manuscript.

The authors performed the requested editorial changes.

Dear Hiroyuki,

Thank you for submitting your revised manuscript to the EMBO Journal. I have now had a chance to take a careful look at everything and all looks good. I am therefore very happy to accept the manuscript for publication here.

Congratulations on a nice study!

Best Karin

Karin Dumstrei, PhD
Senior Editor
The EMBO Journal

Please note that it is EMBO Journal policy for the transcript of the editorial process (containing referee reports and your response letter) to be published as an online supplement to each paper. If you do NOT want this, you will need to inform the Editorial Office via email immediately. More information is available here: http://emboj.embopress.org/about#Transparent_Process

Your manuscript will be processed for publication in the journal by EMBO Press. Manuscripts in the PDF and electronic editions of The EMBO Journal will be copy edited, and you will be provided with page proofs prior to publication. Please note that supplementary information is not included in the proofs.

Should you be planning a Press Release on your article, please get in contact with embojournal@wiley.com as early as possible, in order to coordinate publication and release dates.

If you have any questions, please do not hesitate to call or email the Editorial Office. Thank you for your contribution to The EMBO Journal.

Corresponding Author Name: Hiroyuki Konishi and Hiroshi Kiyama

Journal Submitted to: The EMBO Journal

Manuscript Number: EMBOJ-2020-104464